# Leveraging biogenic resources to achieve global plastic decarbonization by 2050

Elisabeth Van Roijen ✉ & Sabbie A. Miller

There is a rising urgency to decarbonize plastic production given its high carbon footprint and rapid growth in demand. Here, we highlight pathways for carbon uptake and temporary storage (i.e., net-negative greenhouse gas emissions) for plastics on a global scale by 2050. We focus on bio-based plastics and consider potential market replacement, renewable energy integration, and waste management practices. Our analysis reveals that achieving net-negative emissions requires high levels of all three strategies. For example, reaching 60% bio-based plastics still requires 100% renewable energy and 90% recycling, while 40% recycling requires 90% bio-based plastics with 100% renewable energy. Maximizing all three variables could store up to 270 million metric tonnes of carbon dioxide equivalents by 2050. By 2030, annual emissions from plastics could be reduced by 58% compared to current levels by substituting 41% of petroleum-based plastics with bio-based alternatives, transitioning to 100% renewable energy, and recycling 27% of plastics at end-of-life.

The high carbon footprint of plastics, driven by their reliance on fossil-fuels as both a feedstock and energy-source, resulted in roughly 2 gigatonnes (Gt) of carbon dioxide equivalent ($CO_2e$) emissions in 2019[1]. In part contributing to these high cumulative emissions, the rate of increase in plastic production has outpaced all other bulk materials and is expected to double in annual production quantities by 2050[2]. Further, the petrochemical sector continues to hold significant power in terms of economic value, representing 4 trillion United States dollars (USD) in sales in 2019[3]. Due to an increasingly large scale of expected production, plastics are on track to contribute to 15% of the global carbon budget by 2050[4]. To achieve the 2 °C targets set by the Intergovernmental Panel on Climate Change (IPCC), a drastic shift in the way we currently produce and dispose of plastics is urgently needed. It is important to note the interdependence of the petroleum-based compounds industries (e.g., conventional plastics, fertilizers) with fossil-energy industries. Recent estimates have suggested that over 10% of the global oil demand is used for petrochemical products[5]. The compounds used to form plastics are currently inexpensively co-produced with fuels (e.g., gasoline, diesel) and the equipment to produce these compounds is capital intensive[6], which can affect decarbonization efforts for plastics. There has been growing interest in the petrochemical sector to diversify its feedstocks[7]. It has been theorized

that plastics can act as a carbon sink when combining renewable energy, recycling, and fully bio-based or $CO_2$-based feedstocks for plastic production[2,8]. However, the ability to successfully implement the use of renewable energy, alternative feedstocks, and recycling technologies will depend on various factors such as resource availability, energy demand, and the market potential of different plastic technologies. Here, we study the cumulative effects of factors such as electricity source, feedstock source, waste management practice, market potential of various plastics, and biogenic resources necessary on the ability of plastics to offer net-negative greenhouse gas (GHG) emissions.

In this work, we utilize harmonized life cycle assessment (LCA) models[9] to quantify net-negative GHG pathways. Namely, we consider three biodegradable bio-based plastics, polylactic acid (PLA), thermoplastic starch (TPS), and polyhydroxybutyrate (PHB), as well as 6 non-biodegradable bio-based plastics: polyethylene terephthalate (Bio-PET), high density polyethylene (Bio-HDPE), polyvinyl chloride (Bio-PVC), polypropylene (Bio-PP), polyurethane (Bio-PUR), and polytrimethylene terephthalate (Bio-PTT). Biodegradable bio-based plastics are included in this investigation because of their projected use and unique suitability in applications such as food packaging, where contamination may inhibit the ability to effectively recycle these

---

Department of Civil and Environmental Engineering, University of California, Davis, CA, USA. ✉e-mail: evanroijen@ucdavis.edu

materials. The GHG emissions associated with end-of-life (EoL) management of bio-based plastics is also considered for various treatments including: landfill, incineration, composting, anaerobic digestion, thermomechanical recycling, and chemical recycling[10]. To model the impact of implementing various technologies and waste management scenarios, we assess the potential utilization of each plastic type at a global scale. By examining these factors and the percent of energy supplied by renewables, various net-negative GHG emission scenarios are identified as are thresholds beyond which carbon-negative emissions would occur. We further utilize technology readiness levels (TRLs) and a resource availability assessment to determine the level of GHG reductions that could be achieved in the short (by 2030), medium (by 2040), and long term (by 2050) (see details in the Methods).

## Results

### Pathways to net-negative GHG emissions
Our findings show there are over 100,000 scenarios that could lead to net-negative GHG plastics based on the assumptions and variables examined herein. Scenarios were determined by running an algorithm for 11 million total combinations (see Methods for detailed assumptions). Note that this list of net-negative GHG emissions is not exhaustive in that only incremental changes of 10% are used for each variable. A maximum reduction of roughly −1.3 kg $CO_2$e per kg plastic can be achieved using 100% renewable energy, 90% bio-based plastics, and 90% recycling. The results showed a mean value of −0.47 kg $CO_2$e per kg plastic (standard deviation = 0.279), with a minimum of −1.36 kg $CO_2$e per kg and a maximum of −0.1 kg $CO_2$e per kg. There is notable flexibility among the required share of bio-based plastics, the fraction of renewable vs. non-renewable energy, and the type of waste management treatment that can be applied while still reaching net-negative GHG emissions on a global scale. Here, we focus on three scenarios that highlight the minimum amount of renewable energy,

recycling, and bio-based plastics, that are required to reach net-negative emissions globally (highlighted in Fig. 1); a full list of these scenarios can be found in Supplementary Data 1, Sheet 8. We discuss these key thresholds to highlight the degree of flexibility for certain parameters such as waste management (which is noteworthy given the potential difficulty in achieving net-zero energy emissions, 90% recycling rate, and a fully bio-based plastic market simultaneously in a timely manner), as well as highlight areas of inflexibility (e.g., requiring a minimum of 70% renewable energy), to emphasize the need for a swift and significant change in the plastic industry.

### Bio-based market share
We find that on average, 83% of the plastic market needs to be bio-based to offer lower than net-zero emissions. However, a minimum of 60% bio-based plastics can achieve net-negative GHG emissions on a global scale if 100% renewable energy and an 80% recycling rate of plastics is achieved. With more than 60% bio-based plastics on the market, there is more flexibility in the other factors. For example, if an 80% bio-based plastic market is achieved, a 60% recycling rate for non-biodegradable plastics could still lead to net-negative GHG emissions. This outcome is notable given that, despite bio-based plastics being around since the 1850s[11], the growth in bio-based plastics has been slow, amounting to only 1% of the petroleum plastic market. This slow adoption suggests a growth in the bio-based plastic market will likely require pronounced policy mechanisms to drive adoption, such as incentives. Furthermore, there might be several limitations to full adoption of bio-based plastics. For example, some commodity plastics, such as polymethyl methacrylate (PMMA) and PVC, do not have commercialized, fully bio-based alternatives available yet[12,13]. While not limited by technology readiness, some bio-based plastics, such as PHB, are expensive to produce and utilize 1st generation feedstocks, or feedstocks that compete with food production[14]. Not requiring a full

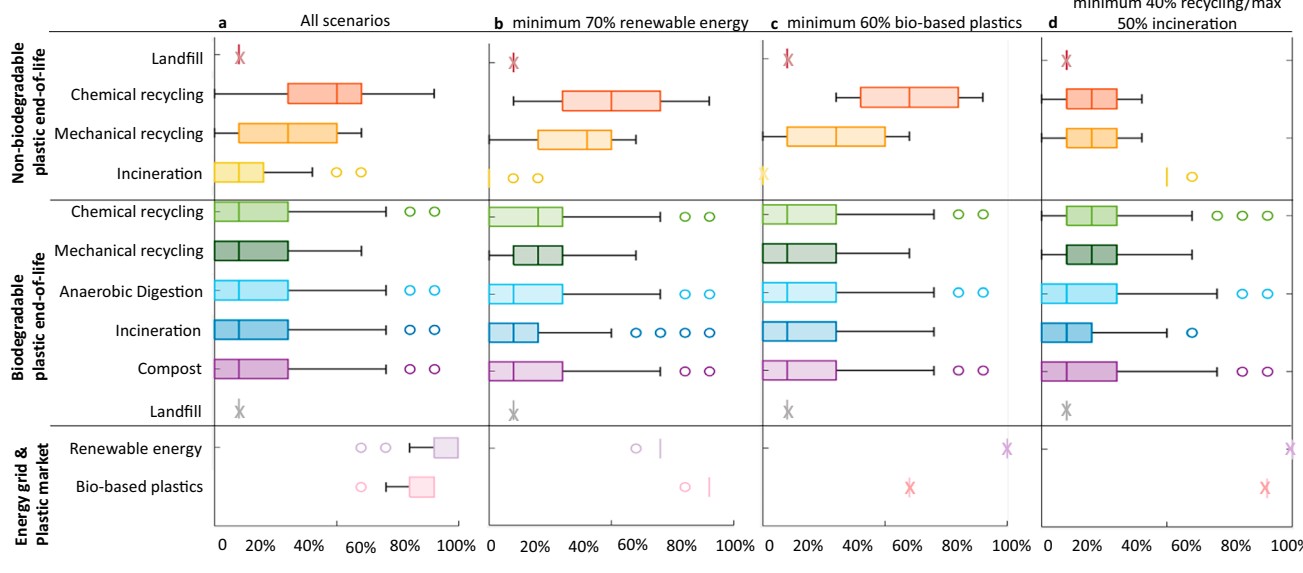

**Fig. 1 | Summary of notable scenarios that lead to net-negative greenhouse gas (GHG) emissions for plastics on a per-kg plastic market basis.** Box plots represent the probability (out of the total number of net-negative GHG emission scenarios) that a given level of implementation of the strategies on the y-axis will lead to net-negative GHG emissions for the plastic industry. Stars indicate single values (e.g., the landfill rate is held constant at 10% for all scenarios). Note that these scenarios are mutually exclusive and cannot be true at the same time. Each panel represents values for each variable that lead to net-negative GHG emissions when (**a**) all scenarios are considered (**b**) the minimum value of 70% renewable energy is used. Note: the outlier for 60% renewable energy reflects 4 scenarios (out of the 100,000) that result in net-negative emissions that require 90% recycling of all plastic and 90% bio-based plastics. (**c**) when the minimum value of 60% bio-based plastics is used, and (**d**) when a minimum of 40% recycling (mechanical + chemical) is used. Note that the outlier for a maximum of 60% incineration reflects a subset of 92 scenarios (out of 100,000) that result in net-negative GHG emissions when 100% renewable energy, 90% bio-based plastics, and a minimum of 80% recycling of biodegradable plastics is achieved. 'Renewable Energy' refers to the percent of the energy demand being sourced from renewable energy vs the 2018 global average electricity grid. 'Bio-based plastics' refers to the percent of the plastic market that is produced using bio-based resources instead of fossil feedstock.

transition to bio-based resources is also notable from a resource availability perspective. Although we find that the current supply of biomass residues is sufficient to meet the demand for some bio-based plastics (see Supplementary Data 1, Sheet 5), a cumulative global shift towards a more bio-based economy is expected to put a strain on resources[15], making it difficult to keep up with plastic demand. Finally, although novel technologies such as $CO_2$ capture and utilization could be utilized as a method for bio-based plastic production, these technologies are not currently viable given factors such as their high cost and energy requirements[16]. Despite this flexibility, even achieving the minimum target of 60% bio-based plastics is going to require key actions from government and industry, such as implementing taxes on traditional fossil-based feedstocks and mandating minimum bio-based content requirements in new products.

## Flexibility of energy grid

Out of all net-negative scenarios, the average renewable energy grid make-up was 93%. A high share of renewable energy is required given that some bio-based plastic production routes, such as PHB from biogas, require significant amounts of energy, and therefore do not necessarily provide a GHG benefit without the use of renewables[17]. However, a minimum of 70% renewable energy can achieve net-negative GHG emission plastics (assuming the rest is satisfied using the 2018 global average electricity grid) if: (1) a minimum recycling rate of 70%; and (2) a minimum of 80% bio-based plastics is achieved. This minimum threshold of 70% renewable energy is pertinent given the notable energy demand for the petrochemical industry (amounting to ~30% of global final industrial energy use, with plastics being its main product)[18]. Compounding this issue, a shift to bio-based plastics may also require high energy demands as a function of agricultural practices. For example, the Haber-Bosch process for ammonia production, which is required for the production of fertilizer, currently utilizes 2% of total energy consumption worldwide[19]. This notable energy demand is partly due to the high temperatures above 700°C and high pressures above 200 bar that are required for such processes[20]. Although some solar thermal technologies can reach temperatures requirements up to 2000 °C[21], solar thermal energy currently only makes up a fraction of the 2.5% of global energy supplied by renewables[22]. Therefore, plastic production may continue to rely on the use of fossil fuels until the global capacity of renewable energy generation can meet the energy demands of the plastic sector and support cultivation of requisite biogenic feedstock resources.

## Flexibility of waste management

On average, 37% of bio-based biodegradable plastics and 77% of bio-based non-biodegradable plastics, need to be recycled in order to reach net-negative emissions globally. It is important to note that even if all other strategies are incorporated, such as 100% renewable energy and fully bio-based plastic markets, we will still need to achieve a minimum of 40% recycling to reach net-negative GHG emission plastics while limiting landfill disposal to 10%. While thermomechanical recycling can be used as a method to effectively recycle thermoplastics, which make up ~85% of the plastic market[23], the global average recycling rate is currently less than 10%[24]. This disparity highlights the need for drastic and immediate changes to waste management systems globally to allow net-negative GHG emissions to be achieved for plastics. Currently, the expense of thermomechanical recycling is unfavorable given the low cost of virgin materials, high cost of manual sorting, and the low-value or degraded performance characteristics of mechanically recycled plastics[25]. Further, the largest growth in plastic waste is expected to occur low-income countries (those with a gross national income of $1135 USD or less in 2024), many of which may not have the requisite waste management infrastructure. Another major roadblock for plastic recycling is contamination: without proper sorting, plastic waste streams are often contaminated with food waste

as well as other materials, making it difficult to effectively recycle. While some new technologies such as chemical recycling can help streamline the waste treatment process by converting mixed plastic waste and contaminated plastics into valuable monomers[26], there is a lack of social acceptance around these technologies outside of the scientific community[27], and in some cases, a lack of established supply chains for low-impact solvents[28]. Therefore, policy efforts such as extended producer responsibility laws[29,30], recycled-content targets[31], or packaging taxes[32], may be necessary to re-shape the global plastic waste management system[33].

Although chemical recycling could help to address some of the issues associated with contaminated and mixed plastic waste streams, it can be an energy- and cost-intensive process, and the technical and economic feasibility of large-scale implementation still needs to be assessed[34]. In this work, we consider the impacts of chemical recycling based on a pyrolysis process for mixed plastic waste. However, we note pyrolysis might not be a viable end-of-life management option for some materials (such as PET or PVC), due to the generation of harmful thermal degradation products[35], and modeling efforts to reflect other chemical recycling routes should be developed in future work.

For biodegradable bio-based plastics, anywhere from 0-90% of the material (assuming 10% is always landfilled or lost to the environment) can be treated with any end-of-life scenario (anaerobic digestion, compost, incineration, recycling, or landfill) and still allow for net-negative GHG emissions globally. The flexibility among biodegradable bio-based plastic waste management options is notable given the technological difficulty associated with bio-based plastic waste treatment as a result of (1) the lack of consumer awareness on proper disposal methods for biodegradable bio-based plastics[36,37] and (2) the lack of adequate separation technology at organic recycling facilities needed to differentiate between biodegradable and non-biodegradable bio-based plastic[38]. For example, it has been argued that the implementation of biodegradable bio-based plastics is favorable for food packaging applications, due to the reduction in microplastic formation as a result of improper disposal (e.g., to the environment), compared to non-biodegradable plastics. Traditional mechanical recycling is not typically suitable for biodegradable bio-based plastics as it can result in significant reductions in material quality. On the other hand, end-of-life treatment options such as composting or anaerobic digestion may be ideal for biodegradable bio-based plastics used in food packaging as it can divert food waste from being landfilled, and reduce methane emissions from uncontrolled landfills[39]. For improved end-of-life management of biodegradable bio-based plastics, identification and separation steps need to be incorporated into commercial composting and anaerobic digestion facilities, and consistent, standardized labeling of bio-based plastics needs to be implemented to improve separation at the source.

## Roadmap for creating carbon uptake in the plastics industry

While the previous sections highlight the minimum thresholds required for recycling, bio-based plastics, and renewable energy, here we provide a roadmap which leverages a combination of these solutions to achieve net-negative emissions by 2050. The combination of solutions presented for the short term (2030), medium term (2040) and long term (2050) were chosen based on TRLs, resource availability, and relevant policies (Fig. 2). TRLs are based on the United States Department of Agriculture definitions[40], and resource availability data are from the Food and Agricultural Organization 2020 statistics[41]. For the near term (2030) scenario, only bio-based plastics with a TRL level of 9 are considered. The 2040 scenario considers bio-based plastics with TRL levels of 5 and above, which encompasses all of the bio-based plastics herein except for bio-PP as it is not considered feasible in the medium term due to resource constraints (see Table 1). While mechanical recycling, composting and anerobic digestion all have TRL levels of 9, we base their

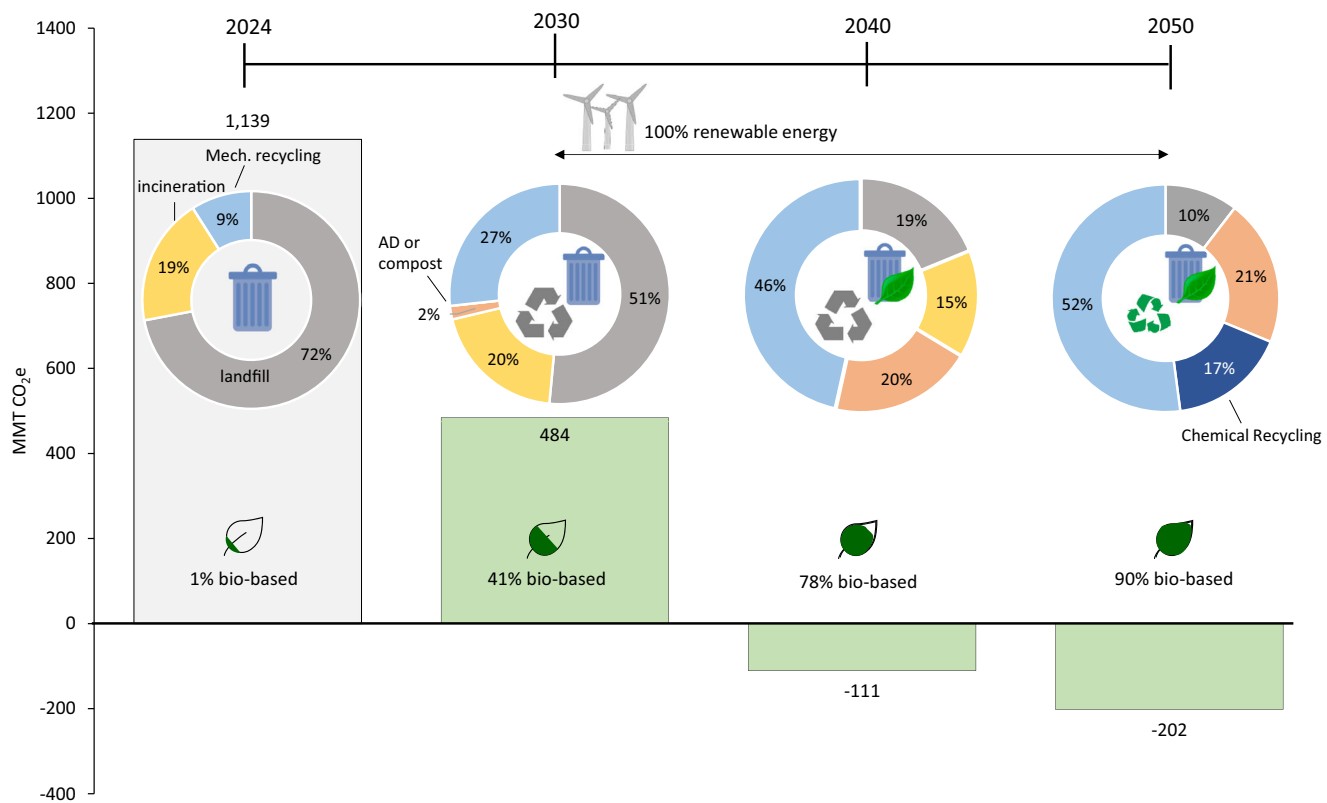

**Fig. 2 | Roadmap for achieving net-negative greenhouse gas emission plastics by 2050, considering both production and end-of-life mitigation strategies.**
Technologies are implemented based on their technology readiness level and resource availability. Source data is available in Supplementary Data 1, Sheet 9.

**Table 1 | Quantity of bio-based resources in million metric tons (MMT) and total energy demand (in EJ) that would be needed to achieve the plastic production scenarios outlined in the roadmap**

| Year | Corn stover | Wheat straw | Biomethane | Used cooking oil | Reclaimed potato starch | Energy demand (EJ) | % of 2019 global energy production (from[110]) |
|------|-------------|-------------|------------|------------------|--------------------------|--------------------|-----------------------------------------------|
| 2030 | 39 | 53 | 0 | 0 | 8 | 9.48 | 1.5 |
| 2040 | 111 | 112 | 42 | 12 | 9 | 8.52 | 1.4 |
| 2050 | 106 | 107 | 40 | 17,395[a] | 9 | 6.30 | 1.0 |

The annual production of bio-based feedstocks is based on 2022 resource data from the Food and Agricultural Organization (FAO); potential changes in future yields of biomass are not considered. The availability of corn stover and wheat straw accounts for the use of some feedstock as a soil amendment. See supplemental data sheets 5 and 12 for detailed calculations for feedstock availability and energy use, respectively.

[a] In this study, we leverage a model for bio-based polypropylene from used vegetable oil which includes an inefficient conversion process of used vegetable oil to bio-based naphtha (wherein the main product is hydrotreated vegetable oil, and only 2% of outputs is bio-based naphtha)[111].

implementation on relevant policies (as described below). Similarly, given the lack of robust infrastructure for collecting, sorting and treating biodegradable plastics, full implementation of composting and anaerobic digestion for these materials is not considered feasible until the medium term (2040). The magnitude of emissions and uptake are calculated based on the assumption that plastic demand will continue to grow at an annual rate of 4% per year. While these strategies do not need to be implemented at this scale or in this order, this analysis suggests that pathways to net-uptake in the plastics industry are feasible within the coming decades.

In the near term (e.g., by 2030), a 58% reduction in global GHG emissions compared to the current petroleum-based plastic sector could be achieved via: 41% substitution of petroleum-based plastic with bio-based plastics, 100% renewable energy, and 27% thermo-mechanical recycling. A 41% bio-based plastic market can be achieved by using PLA, TPS, Bio-PVC and Bio-PE, all of which have full-scale commercial production capabilities (see Supplementary Data 1, Sheet 6). Table 1 includes a summary of the quantity of biomass resources that would be utilized, as well as the total amount of energy required

(in EJ) to meet the roadmap goals. In 2030, a 41% bio-based plastic market is feasible from a resource availability perspective, requiring 39% of available corn stover (e.g., corn stover not used as a soil amendment), 53% of available wheat straw, and roughly 8% of reclaimed potato starch produced globally (see Supplementary Data 1, Sheet 5). While the technologies and resources are available for this replacement rate, European Bioplastics[42] predicts that global bio-based plastic production will only reach 6.3 MMT by 2027, roughly 25 times less than the ~165 MMT of bio-based plastic production required to reach this 2030 target. However, these projections are based on current market behavior. When looking strictly at technological availability, the European Technology platform for sustainable chemistry estimates that as much as 30% of the chemical industry in Europe could be sourced by renewables by 2025[43]. These findings suggest the need for strong policy incentives and/or regulatory actions such as mandating a minimum bio-based carbon content for new products.

Other studies that have looked at decarbonizing the plastic sector, often consider carbon capture and utilization (CCU), which has a high energy demand[44]. In this study, we exclusively consider biomass-based

production routes, which have the potential to reduce the energy demand compared to a fully fossil-based plastic market. In 2030, a 40% bio-based plastic market combined with a 27% recycling rate would require 9.48 EJ of energy, which is a 24% reduction compared to business-as-usual (assuming the average energy-demand for plastic production is 32 MJ per kg).

A thermomechanical recycling rate of 27% could be achieved assuming policy mechanisms are introduced to incentivize recycling and improve the sorting efficiency of plastic waste. This recycling rate aligns with goals on both regional and global scales; (1) in Europe, the plastic industry aims to increase recycling to 25–27% by 2030[45,46], (2) the United States Environmental Protection Agency (US EPA) is targeting a 50% recycling rate of all materials by 2030[47], and (3) the United Nations (UN) recently reported targets for a 20% recycling rate for short-lived plastics by 2030[48]. In 2021, the US EPA introduced the National Recycling Strategy, which aims to improve markets for recycling, increase collection rates, reduce contamination in recycled streams, enhance policies to support circularity, standardize measurements and increase data collection[49]. In addition to supportive policies, further research and development efforts are needed to identify more efficient and cost-effective recycling strategies. For example, the BOTTLE consortium led by the United States Department of Energy investigates new plastic recycling technologies and bio-based plastic production routes at the lab-scale while simultaneously conducting techno-economic assessments and LCAs to concurrently assess the economic viability and environmental impacts of new processes[50].

Globally, a carbon-negative plastics industry can be achieved in the mid-term (e.g., 2040) by increasing the bio-based plastic market share to nearly 80% and by limiting landfilling of all plastics to 20% (with the remainder of plastics going to recycling or composting and anaerobic digestion). This combination would result in −110 MMT of $CO_2$e emissions/yr from the plastic industry in 2040 and a 36% reduction in energy demand compared to business-as-usual fossil-based plastic production. Bio-based plastics that are considered suitable for this midterm goal include Bio-PET, Bio-PTT, Bio-PP, PHB and Bio-PUR, (in addition to the bio-based plastics introduced in 2030) given their proximity to full-scale commercialization. Reaching an 80% bio-based plastic market by 2040 aligns with the United States' goals to commercialize bio-based materials to substitute 90% of today's plastics within 20 years[51]. Shifts towards a bio-based economy are also occurring outside the United States, with over 30 countries developing a national bioeconomy strategy[52]. This level of substitution would require slightly more corn stover and wheat straw than is currently available (111% and 112% respectively), as well as 9% of reclaimed potato starch, 12% of used cooking oil, and roughly 42% of global biomethane production (Table 1). It is important to note that the availability of feedstocks presented in Table 1 is based on 2022 global production values, and the quantity of these agricultural byproducts may increase in the future. For example, studies have projected that annual crop production will have to at least double by 2050 to meet increases in food demand[53]. A recycling rate of 46% by 2040 is in-line with estimates in the literature, such as the Ellen MacArthur Foundation, which calls for a 67% recycling rate by 2040[54], Plastics Europe, which targets a 46% recycling rate by 2040[55], and the UN, which estimates a 56% recycling rate by 2040[56]. The Organization for Economic Co-operation and Development (OECD) modeled in their report on the future global outlook of plastics, that introducing a tax increase for plastic and plastic packaging (rising to 750\$/tonne by 2060) would result in a recycling rate of 40%, which we identify herein as a necessary threshold[57]. A recent study found that four policies could help improve plastic waste management: (1) requiring 40% post-consumer recycled plastic in new products, (2) capping new plastic production levels to 2020 (547 MMT), (3) investing in the expansion of waste management infrastructure (especially in lower-income countries), and (4),

implementing a small fee on plastic packaging[58]. In addition to these potential policies, investments in waste management infrastructure are also needed. To help expand waste management infrastructure, the US EPA is investing \$275 million in solid waste infrastructure for recycling grants which supports projects related to recycling, composting as well as re-manufacturing facilities[51,52,57,59].

Although net-negative GHG emissions are achieved in the mid-term goal, this scenario still results in roughly 15% of plastics being incinerated, 20% being landfilled, and roughly 20% of plastics made from fossil-fuels. In the long-term (2050) scenario, to create a more circular economy, recycling (using renewable energy), alongside a maximum substitution of bio-based plastics (90%), can be employed to further reduce emissions. Maximizing the share of bio-based plastics in 2050 would include the substitution of bio-PP. In this study, we model the production of bio-PP from used vegetable oil which includes an inefficient hydrotreatment conversion process of used vegetable oil to bio-based naphtha. As a result, the quantity of used vegetable oil required would be roughly 174 times the current supply (Table 1). While this production process is common in industry, companies such as Borealis are in the process of expanding their production of bio-PP to meet higher demand[60]. Therefore, it is assumed that alternative and /or more efficient production routes will be available by 2050. Other production routes for bio-PP include (1) traditional fermentation of biomass feedstocks to bioethanol, dehydration to ethylene, subsequent transformation to butene via dimerization, and metathesis of ethylene and butene to form propylene monomer[61]. (2) Gasification of agricultural waste to generate bio-syngas, conversion to dimethyl ether and final conversion to propylene and ethylene[62]. (3) Gasification of biomass to produce syngas, conversion to methanol and subsequent conversion to dimethyl ether to produce propylene and other byproducts. Utilizing a combination of these different production routes can expand the types of biomass residues needed for bio-PP beyond vegetable oil (e.g., woody biomass, agricultural residues etc.). Although certain bio-based plastic production routes (such as PHB from bio-methane) have a high energy demand, on average bio-based plastics can offer a reduction in energy demand compared to fossil-based plastics. As a result, increasing the share of bio-based plastics (along with an increase in recycling rates), results in a 33% reduction in energy demand in the 2050 scenario versus the 2030 scenario, despite increases in plastic demand.

Given the technical readiness of thermomechanical recycling, this value is assumed to be maximized to treat up to 52% of plastic waste by 2050. This value reflects the estimated market share of non-biodegradable thermoplastics (~70%) in 2050 combined with a 75% assumed sorting efficiency. This long-term combination of mechanisms may help minimize adverse environmental impacts associated with landfilled plastics[63], while simultaneously resulting in an annual carbon storage of ~270 MMT and a 50% reduction in energy demand compared to business-as-usual fossil-based plastic production. In addition, it helps to conserve resources and increase the availability of bio-based resources for other applications, which will be increasingly important as the global population grows to 10 billion in 2050, associated with significant increases in crop demand[64]. While it is difficult to make direct comparisons to previous literature due to differences in scope, these results align with findings from Meys et al.[8], where it was found that a combination of 70% recycling, renewable energy, biomass feedstocks and CCU could lead to −30 MMT of $CO_2$ annually.

Recent reports by the IPCC highlight the need for carbon dioxide removal strategies (CDR), on top of strict decarbonization efforts, to stay below a 2 °C global warming target. Namely, a maximum of roughly 1100 Gt of CDR is necessary[65,66]. Increasing the rate of chemical recycling, as seen in the 2050 scenario, would enable these bio-based plastics to act as a CDR strategy, with an annual carbon storage capacity of 270 MMT of $CO_2$.

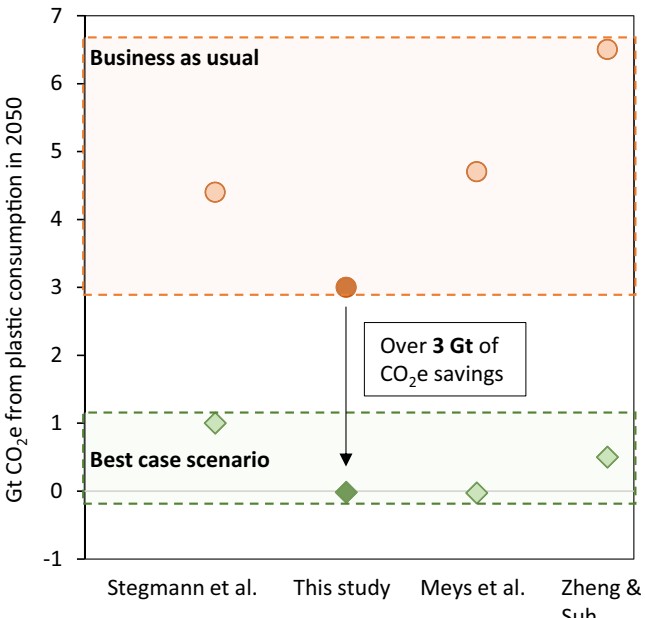

**Fig. 3 | Comparison of roadmap results from this study to future petroleum-based plastic life cycle emissions in a business-as-usual scenario.** For this study, "Best case scenario" refers to a scenario in which 90% of the plastic market is bio-based, 90% of plastic is recycled, 10% is landfilled or lost to the environment, and 100% renewable energy is utilized. For the other studies, "Best case scenario" refers to the scenario that resulted in the lowest possible emissions. The shaded regions show the range of expected emissions under a business-as-usual approach (orange), and a best-case-scenario approach (green).

Note that the results presented in Fig. 2 are based on a 4% annual growth rate in plastic demand. Once net-negative GHG emissions are achieved, the growth in plastic demand is what drives greater magnitudes of carbon storage potential. Therefore, a sensitivity analysis was conducted to examine the resulting GHG emissions for these same scenarios under different growth rate assumptions (namely assuming no growth in plastic demand, a 2.5% annual growth rate, and an 8% annual growth rate). It is important to consider reductions in future plastic demand given the possible impact of various policies such as single-use plastic bans, packaging taxes, or minimum recycled contents, on the demand of virgin plastic production. Further, reducing the growth in plastic demand from 4% to 2% per year can result in a 56–81% reduction in GHG emissions compared to business-as-usual[4]. We found that, even if plastic demand does not increase in the future, negative emissions are still observed at a rate of −59 MMT and −94 MMT $CO_2e$ per year by 2040 and 2050, respectively (see full results in Supplemental Data Sheet 10). Therefore, although increases in plastic demand result in greater levels of $CO_2$ reduction, an increase in demand is not necessary to achieve negative emissions. Further, while this study considers GHG emissions, there are various other factors such as land use and water consumption, that would likely increase with an increase in plastic demand. Therefore, future work should leverage the results of this study to determine scenarios in which these other environmental factors are also minimized. Another important limitation of this study is the simplified assumption that the magnitude and composition of plastic reaching end-of-life is the same as the amount and type of plastic being produced each year. Given that some plastics can have longer lifetimes (such as 35 years in the construction sector), the amount of plastic reaching end-of-life in a given year could be less than the amount produced. This simplified approach was used in this study to provide a conservative estimate for GHG emissions: under low recycling rate scenarios, the emissions from incineration provide a high-end estimate for plastic end-of-life impacts, while under

high recycling-rate scenarios, the increase in plastic recycling reduces the amount of new carbon being stored in the materials, hence leading to lower overall carbon storage potential. Future work should expand upon this study by incorporating a material flow analysis to better model the temporal impacts of plastic production and disposal.

Following the targets for renewable energy, bio-based content, and waste managements highlighted in this roadmap, over 3 Gt of $CO_2e$ could be avoided in 2050 alone compared to a business-as-usual scenario (Fig. 3). Further, this pathway could shift the plastics industry from being 15% of the carbon budget to a $CO_2$ removal strategy. Previous literature has reported the potential for low-carbon or net-negative GHG emission plastics. However, this study fills important research gaps left outside the scope of previous work by (1) incorporating the impacts of $CH_4$ and $N_2O$ emissions associated with bio-based plastic production, (2) examining the utilization of non-edible bio-based feedstocks for plastic production rather than $CO_2$ feedstocks or food products, and (3) incorporating technology readiness levels and resource availability to provide a timeline of feasible mitigation strategies.

## Discussion

In this study, we find that globally, a minimum of 40% recycling (alongside 100% renewable energy and 90% bio-based plastics), or 60% bio-based plastics (alongside 90% recycling and 100% renewable energy) needs to be achieved to reach net-GHG-negative emissions. Further, we see that a combination of strategies, such as 47% recycling, 100% renewable energy, and 80% bio-based plastics, could lead to the plastics industry acting as a carbon sink by 2040. A shift towards a largely bio-based plastic market will be challenging due to the vast current reliance on fossil-fuels as both an energy source and feedstock in the petrochemical sector. Further, given that the plastic market is expected to continue to grow significantly in the coming decades, leading producers have invested heavily in expanding infrastructure for this fossil-fuel industry. While some bio-based plastics could be directly used in existing infrastructure due to identical chemical structures to petroleum-based plastics, there still exists the roadblock of changing feedstock value chains. Currently, the plastic industry is strongly tied with the fossil-fuel industry. For example, oil and chemical companies have joined together to create the world's largest petrochemical facility[67]. Therefore, policy initiatives in the form of tax incentives, bans on certain petroleum-based plastics, and continued investment in research and development, will be needed to overcome existing institutional and political partnerships within the petrochemical sector. Some regions, such as the United States and Europe, have stated goals for the plastic industry that align with the targets set forth in this roadmap. However, most plastic growth is expected to occur in low-income countries with less developed waste management systems[68]. Therefore, policies which enable these countries to take part in the transition to a more circular, bio-based plastic economy are needed. Currently, the UN is developing an international legally binding agreement to help minimize plastic pollution[69]; however, a holistic assessment of both the production and end-of-life impacts of plastics is necessary to achieve climate change mitigation goals.

It is also important to consider the costs and energy demand associated with the transition to a circular, bio-based plastic economy. Although the production of bio-based plastics may result in significant energy demands due to additional pretreatment and processing steps required for the conversion of feedstocks[70], the increased rates of recycling could help drive down energy demand. Stegmann et al., found that a 30% reduction in final energy use, compared to a business-as-usual scenario, by 2050 could be achieved with a recycling rate of more than 70%[2]. In this study, we find that shifting to 41% bio-based plastics and a recycling rate of 27% reduces the energy demand of the plastic sector by 24% compared to a business-as-usual scenario.

Further increasing the bio-based plastic content to 90% and recycling rates to 70% by 2050, results in a 50% reduction in energy demand compared to business-as-usual. Despite the relatively high capital costs of large-scale bio-based plastic production and chemical recycling, continuing to rely on fossil resources and the need to implement carbon capture and storage to meet climate change mitigation goals would lead to overall higher costs compared to a bio-based plastic market[8]. We note the life cycle inventory (LCI) data used in this work relies on lab-scale data for energy consumption and chemical inputs. When these lab-scale processes are optimized for efficiency by improving product yields and chemical recovery, the cradle-to-gate impacts of some bio-based plastics have been found to decrease by 91–97%[71]. Therefore, it is important to consider the impacts of scaling production on expected emissions, as this could enable the bio-based plastic market to generate an even larger carbon sink and reduce final energy demand relative to results we present. Further, creating value chains out of otherwise low-valued materials such as agricultural residues, can help stimulate regional economies: in Europe, the bio-based product sector employs 17.42 million people and generates 684 billion US dollars. Despite making up only 3% of the overall chemical market, bio-based chemicals and plastics alone contribute to 10% of the value added from bio-products, highlighting the potential for significant economic growth[72]. It is also important to note the availability of waste streams for use in the bioeconomy: Europe consumes roughly 58 MMT of plastic annually and simultaneously produces between 118 to 138 MMT of bio-waste which could be leveraged as a feedstock[73].

This study utilizes a bottom-up LCA approach, and some technologies were not considered due to limited data availability. Therefore, future work could be expanded to examine novel bio-based plastics such as bio-based TPS blends (e.g., TPS/PLA), and fully bio-based or iso-cyanate free PUR plastic[74]. In addition, the use of $CO_2$ from flue gas from other industries as a feedstock for plastics, often seen as a solution to the hard-to-abate industrial GHG emissions, could be extremely valuable not only from a $CO_2$-removal perspective but also from a resource conservation perspective, opening up the use of biomass residues for other applications. Furthermore, the LCI data for bio-based plastics used in this study were based on an attributional approach, only looking at the processes directly involved with bio-based plastic direction. However, there are many opportunities for co-product valorization, such as the production of diesel and bio-based plastics from waste vegetable oil[75]. Therefore, future work should expand the system boundaries to consider multi-product biorefineries, which may help lower the cost as well as make the bio-based plastic production process more efficient[76].

In this study, only GHG emissions are examined, but it is important to consider in future analysis the multitude of other environmental impacts that arise from agricultural processes such as eutrophication, acidification, water consumption and land-use change[77]. Bachmann et al.[78] applied the planetary boundaries framework to the plastic sector and found that a climate-optimal scenario for the plastic industry, requiring the use of bio-based plastics and renewable energy, would transgress the safe operating space for biosphere integrity, nitrogen and phosphorous flows, but would stay within a safe operating space for freshwater use and land-system change. Similarly, Zheng and Suh[4] report that a complete shift to bio-based plastics would require 5% of total global arable land. However, these estimates do not consider potential future developments in $CO_2$ or algae-based plastics which could help reduce land requirements. While we did not quantify total energy consumption for the global plastic sector under the net-negative GHG emission scenarios examined herein, we do provide the energy-intensity of each bio-based plastic production route in the supplemental materials (see supplemental data sheet 12). Further, Stegmann et al.[79], found that combining bio-based plastics with increased recycling rates not only lowers GHG emissions but also reduces the final energy demand of the plastic sector.

## Methods

### Life cycle inventories

Cradle-to-gate LCIs for nine different bio-based plastics, including three biodegradable bio-based plastics (PLA, PHB, TPS), and 6 non-biodegradable bio-based plastics (Bio-PET, bio-HDPE, Bio-PVC, bio-PP, bio-PUR, and bio-PTT), are utilized from our prior efforts in deriving harmonized plastics datasets[9], which uses consistent scoping, allocation and impact assessment methods across all materials. The GHG emissions from these inventories, as well as permutations with varying energy grids, are used to represent the projected environmental impacts for bio-based plastics from 1st, 2nd and 3rd generation feedstocks. In this work, 2nd and 3rd generation feedstocks refer to non-edible feedstocks that do not compete with food production and feedstocks with a negligible land footprint, respectively. The cradle-to-gate LCIs of petroleum-based plastics are derived from ecoinvent[80], and we have adapted these inventories to ensure corresponding allocation methods, cut-off rules, energy grids, and other modeling assumptions are the same as the bio-based plastics inventories.

End-of-life impacts for biodegradable bio-based plastics are determined based on our previous work, a literature review of experimental biodegradation studies of bio-based plastics[10]. This review includes maximum biodegradation rates for each of the plastics considered in this work under varying disposal conditions (landfill, anaerobic biodegradation, composting). Given that the biodegradation rate of biodegradable plastics varies greatly depending on environmental factors (such as temperature, moisture content, and oxygen availability), as well as material characteristics (such as material thickness and surface area), these values are analyzed using a Monte Carlo simulation ($n = 1000$) to capture the range of potential emissions that can occur. Namely, the maximum level of biodegradation observed within a year for each plastic under different conditions was varied using normal distributions. Due to the lack of data surrounding the behavior of TPS in landfill conditions, biodegradation data from anaerobic digestion studies are used. The median values resulting from these distributions are then used as inputs to the model to determine the lifecycle impacts of biodegradable materials. An attributional approach is utilized to determine the environmental impacts for end-of-life treatment, in line with cradle-to-gate impact assessment methodology used herein. No 'credits' (i.e., negative $CO_2$ emissions) are applied for the generation of energy from incineration or anaerobic digestion (AD), given that the share of renewable energy is expected to continue to increase in the future. Similarly, given the uncertainty of the impact of bio-based plastics on resulting compost quality, no credits are applied for the generation of fertilizer for the anaerobic digestion or composting scenarios.

Certain modeling assumptions were imposed for end-of-life management pathways to ensure reliable results. On average, roughly 70% of global landfilled waste is currently sent to an uncontrolled landfill or disposal environment without biogas capture[81]. The remaining 30% go to a landfill with biogas capture and flaring. This approximate statistic aligns with more detailed monitoring at national levels. For example, according to the US EPA, only 500 out of the 2500 landfills that are monitored nationally currently have landfill gas capture technology[82]. Therefore, in this model, it is assumed that roughly 80% of landfilled plastics are sent to a landfill without biogas capture. The make-up of thermophilic (high temperature) and mesophilic (low temperature) anaerobic digestion systems were assumed to be the same as the European average, 43% and 57%, respectively[83]. These values were chosen due to the availability of data and the prevalence of anaerobic digestion in this region. In general, it is conservative to assume a lower proportion of thermophilic relative to mesophilic anaerobic digesters globally given their higher operating costs[84]. The LCI for thermomechanical recycling of all plastic (both bio-based and petroleum-based) is assumed to be the same and is extracted from existing datasets[80] to remain consistent with other LCIs modeled. The LCI for chemical recycling was obtained from Jeswani et al., which

provides data for the pyrolysis process for mixed plastic waste and supported a consistent modeling effort[85]. This pyrolysis inventory includes energy requirements for pyrolysis but does not include treatment and refining of oil to produce new polymers. A cut-off approach is utilized for both recycling inventories, wherein only the impacts of plastic waste treatment is included (no downstream processing of recyclates, or credits for replacement of virgin plastic is considered).

## Identifying carbon dioxide removal pathways

The variables considered include: (a) the fraction of plastic coming from bio-based resources; (b) the fraction of energy coming from renewable resources (modeled herein as electricity being satisfied by wind, and heat being satisfied by bioenergy); (c) the distribution of waste management options for biodegradable plastics; and (d) the distribution of waste management options for non-biodegradable plastics. Waste management options for biodegradable plastics include: (a) composting; (b) incineration; (c) anaerobic digestion; (d) landfilling; (e) thermomechanical recycling; and (f) chemical recycling. For non-biodegradable plastics, the waste management options at end-of-life include: (a) incineration; (b) landfill; (c) thermomechanical recycling; and (d) chemical recycling.

## Production and end-of-life impacts of bio-based plastics

Cradle-to-gate impacts for bio-based plastics are derived from a comprehensive bottom-up LCA study that includes detailed inventories and results for bio-based plastics capable of replacing 80% of the current plastic market[9]. This study examined 1st, 2nd, and 3rd generation feedstocks for bio-based plastic production. The scope of analysis included direct land-use change emissions associated with growing crops, agricultural processes such as fertilizer application and maize drying, as well as conversion of biomass feedstock into polymers. A mass allocation method was applied to divide up the impacts of upstream agricultural processes between the agricultural feedstocks (e.g., corn vs corn stover). The main bio-based plastic production routes examined include: fermentation of simple sugars, such as sugarcane molasses, as well as lignocellulosic feedstocks, such as corn stover and wheat straw into ethanol (for further conversion to bio-based PET, HDPE, and PVC); production of PHB from biogas (methane) via nutrient limitation of microbes; pretreatment and fermentation of corn stover to lactic acid (for further conversion to PLA); enzymatic hydrolysis and hydrogenolysis of corn stover to form 1,3-propanediol (PDO) (for further conversion to bio-based PTT); conversion of rapeseed oil or used cooking oil to bio-based polyol via amidization with diethanolamine (for further conversion to bio-based PUR); and pretreatment and de-oxygenation of used vegetable oil, producing bio-based naphtha, which is converted to propylene via steam cracking (for further conversion to bio-based PP). More details regarding each of these conversion pathways can be found in our previous work[88]. It is important to note that more efficient biomass conversion processes may be feasible in the future, leading to lower overall GHG emissions and energy consumption.

When examining the total possible net-negative GHG emission pathways, an average value for the cradle-to-gate impacts for 2nd and 3rd generation feedstocks for each bio-based plastic was used. For PHB from landfill biogas, carbon credits were applied assuming that 50% is in the form of avoided methane emissions (e.g., a global warming credit of 28 kg $CO_2$e per kg), and 50% is in the form of avoided $CO_2$ emissions. Although the majority (80%) of landfills do not have gas capture technology in place, we make a conversative estimate for possible avoided methane given that other sources of biogas that could be utilized for PHB generation, such as wastewater treatment facilities, anaerobic digesters, or natural gas plants, all of which may have gas capture in place. First generation feedstocks were not considered when determining net-negative GHG emission pathways given the need to limit competition with food in the long-term. However, when investigating near-term solutions for the roadmap, impacts from

1st generation bio-based plastics are considered based on their technology readiness. See Supplementary Data 1, Sheet 3 for a full list of production impacts of bio-based plastics with and without renewable energy. Given that this dataset only covers 80% of the bio-based plastic market, the remainder (namely cellulose films, Bio-polyamide (PA), Bio-acrylonitrile butadiene styrene (ABS), Bio-polybutylene (PB), Bio-polyacrylates, and Bio-epoxy resins) are assumed to have impacts equal to the average of the other bio-based plastics.

## Assumptions

Only scenarios leading to overall GHG emissions below −0.1 kg $CO_2$e per kg were considered. This value was chosen to limit results to scenarios that contribute to relatively notable carbon uptake, and to conservatively capture effects of uncertainty. A maximum recycling rate of 90% is assumed. This assumption was based on an expectation that there will always inherently be some losses to the environment/landfills. We note this assumption is a slightly more conservative estimate than what has been assumed in previous studies examining the carbon uptake potential of plastics, which have noted higher recycling rates[8].

Based on the mechanical performance characteristics of bio-based plastics, they are currently capable of replacing 90% of the petroleum-based plastic market[86]. See Supplementary Data 1, Sheet 2 for a detailed breakdown of the substitution capability of various bio-based plastics. A limitation of 90% market replacement is partially due to the fact that fully bio-based alternatives have not yet been identified or have remained in the lab-scale for some plastics, including polystyrene (PS), polycarbonate (PC), and polymethyl methacrylate (PMMA), as well as some important chemical building blocks such as toluene di-isocyanate (TDI) and methylene diphenyl di-isocyanate (MDI), which are important inputs for bio-based PUR. Partial bio-based alternatives have emerged in the market for some of these materials (e.g., Mitsubishi started producing partially bio-based PC from biomass[87]) therefore it is possible that, upon further research and development efforts, more of the plastic market could be bio-based, but a conservative estimate is made herein assuming a maximum of 90% bio-based plastics.

Renewable energy scenarios are modeled using inventories for wind and biogas to cover all electricity and heat requirements (as modeled in ref. [88]). Alternatively, the non-renewable energy scenario uses an electricity grid modeled after the 2018 global average (see Supplementary Data 1, Sheet 10). For plastics, only processes that contributed to at least 5% of overall GHG emissions when using the original inventories are assumed to be met by renewables. Processes that contributed less than 5% of total GHG emissions were modeled with the original energy resources. For example, if the construction of the chemical plant has a marginal impact of less than 1% of the overall GHG emissions of a bio-based plastic production process, the energy-grid is not changed within the LCI for the chemical plant construction.

The model for "landfill" as waste management option is considered to also capture any plastic losses to the environment. Further, it is assumed that the rate at which plastics were sent to landfills for both biodegradable and non-biodegradable plastics would not be lower than 10%. This 10% is implemented to reflect inherent losses that are assumed to occur either via losses along the waste management supply chain process, or directly from microplastic sources that are hard to control such as tire wear and personal care products[89]. Furthermore, a landfilling rate greater than 10% is not investigated in order to focus only on net-negative GHG emission pathways that promote a more circular economy and reduce the strain on biomass resources.

It was assumed that the maximum thermomechanical recycling rate for both biodegradable and non-biodegradable plastics was 60%. A 60% cap on thermomechanical recycling is reflective of the fact that only 85% of the 2023 plastic market is made up of thermoplastics that are suitable for thermomechanical recycling[90,91], and that there is an assumed sorting efficiency of 75%[8,92] given the prevalence of contaminants and mixed plastic waste streams. As such, to reach 90%

recycling rates, the remainder of recycling is achieved via chemical recycling. Although chemical recycling can currently be energy-intensive and expensive, scenarios are examined wherein up to 90% of plastic is fully treated via chemical recycling.

### Roadmap for net-negative GHG emission plastics

It is assumed that the demand for plastics is going to continue to increase at a rate of 4%/year from now to 2050. We also assume that the quantity of plastics reaching end-of-life each year is equal to the quantity of plastics being produced. Although there are plastics that are used in long-term applications such as construction and transportation, we made this assumption to have a "worst-case scenario" estimate for end-of-life emissions. It is important to note that accounting for the longer use-phase of plastics, especially when bio-based, could lead to lower overall GHG emissions. It is also assumed that energy requirements for plastic production can be satisfied by renewables in the near term (2030), based on technology readiness.

### Bio-based plastic market assumptions

The implementation of bio-based plastics within the roadmap was determined based on TRLs as well as resource availability. Bio-based plastics that are currently at a TRL 9 and being produced at full scale include PLA, Bio-PE, TPS and Bio-PVC. For example, NatureWorks, the largest producer of PLA, has a capacity of 150,000 metric tons (t) and is in the process of constructing an additional manufacturing plant in Thailand, which would expand capacity by an additional 75,000 t[93]. Other major PLA producing companies include TotalEnergies Corbion (capacity of 75,000 t)[94] and Futerro (current capacity of 100,000 t)[95]. Major producers of TPS blends include Novamont (capacity of 110,000 t)[96], Kuraray[96], and Biotec (capacity of 30,000 t)[97], with products containing a bio-based content of 40-69%. It is important to note that these companies currently utilize 1st generation crops, such as corn and sugarcane, in their production processes. However, we find that even if all PLA and TPS were produced with corn, it would amount to a demand of less than 4% of global annual corn production (See Supplementary Data 1, Sheet 5 for full resource availability results). Finally, various petrochemical companies have expanded their production line to include bio-based polyethylene such as Braskem[98] and Borealis[99], and Bio-based PVC, such as Ineos[100]. Although these materials were traditionally produced using sugarcane, Borealis and Ineos utilize waste vegetable oil or corn stover as feedstocks for production.

Bio-based plastics that are only produced on a prototype/field scale or have limited feedstock availability are assumed to be at TRL range of 5–7. In the roadmap, we consider these plastics to be implementable by the midterm (2040). Fully bio-based PET materials are currently being produced on a prototype scale (such as Coca-Cola plant bottle[101]). Although PHB is currently being produced on a commercial scale[102], most companies rely on sugar from sugarcane as a feedstock (a 1st generation feedstock). To reach full-scale production for PHB that would satisfy 11% of the global plastic market, this feedstock pathway would require using 42% of globally available sugar production, which would result in significant competition with food production. However, companies producing PHB from waste (such as Mango Materials[103] which makes PHB from waste biomethane), are in the process of scaling up production. Bio-based polyol, a key component for Bio-PUR, is currently being produced on a commercial scale but is typically made from vegetable oil, or corn, which would have to compete with food resources to meet PUR demand[104]. Similarly, Bio-PP is currently being made on a commercial scale from waste vegetable oil[60], but the availability of resources is roughly 200 times lower than necessary to meet PP demand. However, it is expected that with proper research and development efforts, barriers such as these will be better understood and potentially overcome by 2040.

Bio-based plastics that are still being developed on the lab scale are assigned a TRL in the range of 1–4, and they are assumed to be implementable by 2050. This set of plastics includes Bio-PTT, Bio-PUR, PLA, TPS and Bio-PET from 2nd generation feedstocks. Furthermore, bio-based plastics that lack sufficient publicly available LCI data, such as fully bio-based PA and bio-based ABS, are also assumed to be implementable in 2050. See supplementary Data 1, Sheets 5 and 6 for resource availability results and TRL's of the bio-based plastics.

### End-of-life assumptions

Given that biodegradable bio-based plastics currently only make up a fraction of the 1% of the plastic market that is composed of bio-based plastics, composting and anaerobic digestion facilities do not have systems in place to differentiate these materials from their non-biodegradable counterparts. Furthermore, given the small volume of biodegradable, bio-based plastics, consumers are not aware of how to dispose of them due to lack of exposure. Therefore, it is assumed that composting and anaerobic digestion will only become suitable by 2040 when biodegradable plastics make up 22% of the global plastic market (based on the above stipulated assumptions in this roadmap), amounting to roughly 170 MMT of biodegradable plastics entering the waste stream.

Chemical recycling is currently being used by some companies for treatment of individual plastics such as PLA[105]. Additionally, some companies such as BASF are investing in the development of chemical recycling processes that handle mixed plastic waste[106]. However, given the high-cost and small-scale use of current chemical recycling systems, it is assumed to be implemented only in the long term (2050) to help minimize landfill rates.

### Sensitivity to chemical recycling methods

Chemical recycling methods need to be leveraged to reach global recycling targets. However, given the lack of large-scale data availability for chemical recycling processes, and the sensitivity of impacts based on the plastic feedstock stream, we conduct a sensitivity analysis to examine how a change in the impacts of chemical recycling via pyrolysis may impact results. We find that when using data published by Xayachak et al.[107], treating 30% of plastics via chemical recycling in 2050 would lead to an annual carbon sink of 0.5 MMT, a nearly 100% reduction is $CO_2e$ storage compared to the baseline scenario (presented in Fig. 2) using data from Jeswani et al. (See Supplementary Data 1, Sheet 7 for detailed inventories and results). Differences in results between these two pyrolysis processes can be partially explained by having different LCI sources (Jeswani et al. use data from a technology developer, whereas Xayachak et al. used an average from data collected in literature), as well as differences in plastic feedstock stream (Jeswani et al. model pyrolysis using a feedstock mix of PE, PP and PS, while Xayachak et al. use an equal mix of PP, HDPE and LDPE). Additional chemical recycling methods such as hydrolysis, glycolysis and hydrogenolysis were also examined and found to have environmental impacts of 7–18 kg $CO_2e$ per kg (or 2–5 kg $CO_2e$ per kg when using renewable energy)[108]. Therefore, without accounting for the avoided burdens of virgin plastic production, these end-of-life methods result in greater GHG emissions than incineration. These high environmental impacts are partially due to limitations of lab-scale studies, which are often not optimized for efficiency[109]. These results highlight the need for a better understanding of the sensitivity of the global carbon footprint of plastics to various chemical recycling technologies. Further, LCA studies of chemical recycling pathways should report results without the application of credits given that the credits will not apply if we shift towards a bio-based plastic economy (e.g., applying a carbon credit for chemically recycling PET for avoiding the production of fossil-based ethylene glycol will not be applicable when ethylene glycol becomes bio-based).

## Data availability

The data generated in this study are provided in the Supplementary Information/Source Data file

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

## Acknowledgements

E.V. gratefully acknowledges support from the Environmental Research Education Fund. S.A.M. gratefully acknowledges support from the National Science Foundation (CBET-2143981). This work represents the views of the authors, not necessarily those of the funder.

## Author contributions

E.V. and S.A.M. both contributed to the data curation, writing the original draft, review and funding acquisition. E.V. contributed to the formal analysis and visualizations. S.A.M. contributed to the conceptualization of the research and supervision.

## Competing interests

The authors declare no competing interests.
