## [Transparent Peer Review file · Nature Communications]

Leveraging biogenic resources to achieve global plastic decarbonization by 2050

Corresponding Author: Dr Elisabeth Van Roijen

Version 0:

Reviewer comments:

Reviewer #1

(Remarks to the Author)

The paper provides an extensive analysis of CDR in plastics, or more specifically, bioplastics. It outlines specific scenarios and thresholds for these factors to make plastics a net carbon sink. As such, the authors present a roadmap detailing the technological readiness and resource availability required to achieve CDR for plastics by 2050. The paper employs life cycle assessments (LCA) to examine emissions reduction potential in different production/products and waste management scenarios

However, the manuscript requires further revisions and improvements before it can be considered for publication. Specific areas need clearer articulation, including the manuscript's technical details, assumptions, and policy implications. Detailed comments and suggested revisions are included below, referencing pages in the manuscript to help guide the authors in refining their work.

Major suggestions:

- I suggest strengthening the conclusions to emphasize the societal and policy implications, as this could broaden the paper's impact and overall appeal. Also, I suggest providing more insight into the specific set of actions needed from industries and governments, which could improve the roadmap's importance in meeting climate goals.
- I also recommend expanding the discussion on technological and policy feasibility. This could be achieved by adding more detail on the economic, regulatory, and technical challenges associated with the proposed recycling and renewable energy targets. Other suggestions would be to discuss potential policy interventions that could make the transition to bio-based plastics and increase recycling rates appear more achievable, affordable, and/or feasible in the future.
- To provide a comprehensive analysis, the paper could expand the discussions of even the LCA to include other environmental impacts like water use, eutrophication, and land use changes. Addressing these alongside GHG emissions would provide a holistic perspective on bio-based plastic production's environmental sustainability. It could also provide more insight into the trade-offs and synergies between waste management strategies and the role of conventional vs degradable bioplastics.
- There are some concerns about the terminologies used in the paper, more specifically related to misuse and lack of consistency. These need to be reviewed and adjusted accordingly.
- I also recommend the study expand its discussion to provide clearer understanding of the technological options included in the analysis. Not only that, but they should include the potential impacts of other options not mentioned directly in this study. Most notably, the paper overlooks the role of demand changes and material substitution on the market size and, therefore, of emissions and removal potential. Also, additional routes, such as refining and processing of bio-based feedstocks, are not discussed in detail or acknowledged. This would improve the overall understanding and limitations of the paper.

Minor Suggestions and Corrections

Abstract: I believe the "100,000 combinations" claim is unnecessary in the abstract, as it is unclear at this point which combinations they are and what they involve. I recommend focusing on the main takeaway messages, such as the potential and overview of the main strategies.

P2: Provide a broader overview of fossil systems. The authors acknowledge the high carbon footprint of plastics but neglect to highlight the relationship between the non-energy use of fossil fuels and the dependency this could lead to overall fossil energy. Even though the non-energy use share of crude oil can increase in the future, it is unlikely to reach very high values (above 60%), which means that plastic demand alone can determine a certain level of fossil energy production in the future. This topic is already highlighted in previous works and could improve the introduction of this paper.

P2: 100% recycling is not necessary. Other studies have shown that bio-based plastics, for instance, can store carbon in

long-lived applications or long-term end-of-life (such as construction materials). Additionally, conventional bioplastics can store carbon by itself if not directly incinerated or recycled (likely leading to uncontrolled end-of-life, which does not deal directly with pollution and other environmental issues, such as microplastics). Finally, other LCA studies have also shown that increased recycling rates might add emissions to the bioplastic chain, which is unlikely to happen for fossil plastics. Based on the abovementioned issues, I'm not convinced that "100% recycling" is needed to achieve CDR potential. In fact, you confirm this in the next section, when a maximum reduction was achieved with a 90% recycling rate.

P3: Some sections discuss the flexibility allowed within scenarios for achieving CDR. More explicit explanations of this flexibility (e.g., how different energy or recycling thresholds affect the outcomes) would make the results easier to interpret.

P4: Besides bioplastic share, which parameter/aspect was the most challenging (or even the determining factor) for not achieving negative emissions?

P4: Expand on biomass availability and resource limitations. This section has a very brief discussion about potential limitations, but it is too broad and general. I recommend adding more specific comments, including quantitative aspects, to this discussion, which is a very important concern. Additionally, this section would welcome a more detailed discussion on technology availability concerns, such as the one originally made for one example.

P4-5: Explain the role of biodegradable plastics in CDR. It is unclear how biodegradable plastics help as a CDR option. How are they able to achieve negative emissions by itself? I understand they are a very important mitigation option for conventional plastic production and end-of-life emissions.

P4-5: Ensure consistency in terminology (1). The previous comment makes me question the usage of the term "CDR" in this paper, which uses CDR in the same as achieving global negative emissions in this sector. CDRs are options that remove carbon by themselves and, hopefully, account for a (total) net negative contribution when including all remaining positive emissions. Achieving net negative emissions does not mean that CDR is being achieved, in fact, we need CDR options to be deployed much sooner to achieve net negative emissions (or even net zero, for that matter). Please note that the two are not the same, this should be improved, and their use must be explained in the manuscript.

P4-5: Ensure consistency in terminology (2). The terms "bio-based", "bioplastic", and "biodegradable" are used in various contexts. Although the terminology is used correctly throughout the manuscript, a clearer distinction between these terms throughout the paper could prevent confusion for new readers, particularly in sections discussing waste management.

P5: Roadmap introduction. Please note that the "large number of scenarios" was evaluated in this study, so make it more specific at the beginning of this section. From a broad context, several other scenarios could be generated from a more integrated and holistic approach, which is not the focus of this paper.

P6-7: Expand on Assumptions in the Roadmap. The roadmap relies on assumptions (like a 4% annual increase in plastic demand and specific recycling rates), but it was not clear from which set of scenarios (and with which criteria they were designed). Discussing the potential variability in these assumptions and their impact on the roadmap's feasibility could add depth to the analysis for decision/policy makers.

P6: "Although CDR is achieved in the midterm goal". Once again, it seems the authors are using the term CDR as the same as achieving net negative emissions. This is not accurate and needs to be fixed.

P6: Show more comprehensive estimates on net negative emissions or total CDR. The authors focus on one set of assumptions for the roadmap, which was already mentioned it is not clear how it was determined amongst the 100,000 scenarios. Beyond this, what is the range of negative emissions achieved in 2050 by all scenarios? What are the mean value and percentiles? Finally, how do these values compare to Integrated Assessment Model (IAM) results for the same variable and time horizon?

P8: "In this study, we find that globally, a minimum of 40% recycling, 70% renewable energy, or 60% bio-284 based plastics needs to be achieved to reach net-GHG-negative emissions" In here the authors use the terminology correctly, for instance. This is not the same as CDR. In fact, more CDR is needed since you still have remaining positive emissions.

P8: Expand on the role of regional policies. It is not clear why the authors claim that policies at the regional level are not sufficient and why this is related to the fact that most of the demand is growing in developing countries. The connection is superfluous and generic, please expand and justify your comment.

P8-10: Address Potential Overlaps in Data Sources. In the LCA sections, some datasets for bio-based plastics appear based on similar assumptions or overlapping studies. Including a brief explanation of how these datasets complement or differ from one another would improve transparency.

P11-12: Include More Data on Specific Feedstock Availability. The discussion on bio-based feedstock availability should be expanded with quantitative estimates. This would give readers a better sense of the practicality of meeting bio-based plastic targets based on global resource limits and sustainability concerns about the supply chain. Additionally, how can land-use change, and agricultural (AFOLU) emissions affect the GHG emissions results from plastics? If accounted in other sectors, due to inventory guidelines, how would this affect overall achievement of net negative emissions?

P13: Role of multi-product routes. The study focused on plastic production options, which, from my understanding, does not include technological options that are able to make a wider range of products (including energy and non-energy products) from biobased sources. These could have the benefit of decarbonizing one sector while simultaneously decoupling fossil dependency for chemical feedstock. This should be acknowledged, as they were the focus of recent articles in the same journal.

Reviewer #2

(Remarks to the Author)

The paper applies a LCA-based method to calculate pathways for plastics production that lead to carbon dioxide removal as a strategy for climate change mitigation. It considers several biodegradable and non-biodegradable alternatives for fossil-based plastics as well as their production, EoL emissions, feedstock availability, renewable energy use, and market penetration to understand under which conditions the global plastic industry would become net-negative. The topic is interesting and timely to support climate policy for the plastics production industry.

First of all, I would like to thank the authors for the easy and interesting read. The data provided allows for validation of results and is relevant to support future studies. I have a few comments below before accepting it for publication, which I hope are useful to improve the key argument made in the paper.

Major:

- 1) A key limitation of the study concerns the temporal dynamics of plastic production, use, and waste generation, i.e. availability for chemical or mechanical recycling. The authors assume that “the quantity of plastics reaching end-of-life each year is equal to the quantity of plastics being produced”, “to have a “worst-case scenario” estimate for end-of-life emissions.” On one hand, it is a worst case scenario for emissions, but on the other hand, it is the best case scenario for plastic waste availability to be recycled. How does this affect current results and how this could be addressed in future studies?
- 2) What precisely makes bio-based non-biodegradable plastics become a carbon sink? Previous studies emphasize CDR role of bioplastics in long-lived applications, but this does not seem to be the case here. Is it the continuous recirculation of biogenic carbon through mechanical/chemical recycling?
- 3) Assumptions about future demand rely on the historical annual 4% growth. At the same time, demand increase is precisely what drives CDR and net negative emissions in the plastics industry in this study. How would the results change considering lower rates of demand increase (e.g., due to bans on single-use plastic taking off)?
- 4) Adding to the comment above, circularity is mentioned more than once in the manuscript. However, only lower ranked R-strategies are being discussed in the study (recycling). Shouldn't a broader circular bioeconomy strategy for the plastics industry also consider reducing plastic demand and extending the lifetimes of plastic products?
- 5) Figure 1: That is an interesting figure but could be improved for readability. Consider: 1) moving the letters a, b, c, d to the upper left side of each panel; 2) bringing non-biodegradable/biodegradable to the left of the labels in the y axis (I'm assuming there are valid for all the panels?); 3) in general, labelling the three main categories there would facilitate for the reader, there is a lot of info in that figure; 4) why in scenario A there is an outlier with 60% renewable energy? Similarly, why is there an outlier on 60% incineration in scenario C? 5) which year is this figure referring to? Or is static considering current production?
- 6) On the roadmap: the authors presented net emissions until 2050. What does it mean in terms of renewable energy use (for plastics production and recycling) and feedstock use in EJ/yr? What about land-use? These values would help clarifying the role of the sector in competition with others for renewable resources globally.
- 7) The main message that I take from this study, given the limitations on future plastic demand, uses and their temporal dynamics, is that the key to net-negative plastics sector lies on producing more plastics, transitioning towards bio-based feedstocks, and improving collection, sorting and recycling methods. If we aim for the thresholds mentioned regarding renewable energy, recycling and feedstock, the plastics industry will reach net-negative emissions. Is that the main message of the paper?
- 8) I think the discussion section can be much improved. The data collection and modelling in the micro-/technology level is impressive and very comprehensive, but I am missing a discussion that connects these results with global climate policy. For instance, what these results mean for global fossil fuels phase-out? What are the implications for achieving global climate agreements?

Minor:

- L48: There's a 'that' missing before 'to achieve net-zero emissions'
L50: 58% reduction relative to which scenario/year? The whole sentence is a bit confusing, consider rewriting.
L201: Same as above (why the difference in 57% and 58% reductions?)
L217: 25-58%? Is this correct?
L416: 0.1 kgCO₂e/kg (kg before CO₂ is missing)
SI file:
L66: bio-based feedstocks?

Version 1:

Reviewer comments:

Reviewer #2

(Remarks to the Author)

Thank you to the authors for the revisions, which have significantly improved the readability, content, overview of limitations, and overall understanding of the paper. The manuscript is much stronger than the initial version, thanks to the additional information provided. However, I still have a few minor corrections and clarifications before it can be accepted for publication. Below, I refer to pages and lines as indicated in the document with track changes.

1. Table 1 is a valuable addition, but it is not referenced in the text. It would be useful to briefly discuss the values for cooking oil beyond what is mentioned in P17L22-23.

While the benefits of co-production are acknowledged, the scale of production raises feasibility concerns regarding the proposed roadmap. Consider discussing alternatives to cooking oil/rapeseed oil for Bio-PP or other strategies to enhance feasibility. For instance, how much of other fuels are co-produced, and how does this compare to the SAF demand expected for 2030/2050?

2. The study considers wind electricity and biogas as renewable energy sources. Table 1 appears to focus on bio-based feedstocks, but total electricity demand is unclear. Many studies emphasize the increasing electricity requirements for net-zero plastics/chemicals, particularly for CCU pathways. For example, Kätelhön et al. (2019) (<https://doi.org/10.1073/pnas.1821029116>) estimates that shifting chemical feedstocks to carbon capture and utilization would require low-carbon electricity equivalent to 55% of globally projected generation in 2030 (not just renewables!). How does this study compare, given its focus on bio-based alternatives?

3. The criteria for selecting net-negative GHG scenarios (based on LCI and kg plastic) and their alignment with the global roadmap remain unclear. The authors mention TRL as a proxy for short-, medium-, and long-term implementation, as well as resource availability (P7L8) and an assumption of 100% renewable energy (P16L38). A sensitivity analysis was added for demand, yet the decision-making process behind scenario selection is still not entirely transparent or scattered in multiple subsections. Moreover, the first sentence of the discussion also mention the minimum shares of recycling, renewable energy, and bio-based feedstock for net negative emissions in plastics (per kg of plastics). The next sentence mention the required shares for the roadmap, which give me an indication that those are the minimum shares for net negative, considering TRL and resource constraints. However, the supplementary material (SDS11) also includes a column labeled "carbon optimal pathway." While this terminology is used in other studies, there is no clear indication of an optimization method applied to determine the carbon-optimal long-term pathway in this study. Is it the minimum effort for net-negative global emissions? Maximum cumulative net negative emissions? Consider adding a sentence, a scheme, or short paragraph that summarizes the steps taken to define the roadmap.

Thank you again for the opportunity of reading and reviewing this paper. I hope these comments are useful to enhance its clarity.

Title: Using biogenic resources to make plastics a global carbon sink: a roadmap for sustainable decarbonization by 2050

Manuscript ID: NCOMMS-24-58695

Response to Reviewer Comments

The authors would like to thank the reviewers for taking the time to provide thoughtful feedback on our work. We have provided point-by-point responses to the comments presented below. Please note that the section and line numbers in our responses refer to the revised manuscript with tracked changes.

Reviewer #1

Comment R1.0. The paper provides an extensive analysis of CDR in plastics, or more specifically, bioplastics. It outlines specific scenarios and thresholds for these factors to make plastics a net carbon sink. As such, the authors present a roadmap detailing the technological readiness and resource availability required to achieve CDR for plastics by 2050. The paper employs life cycle assessments (LCA) to examine emissions reduction potential in different production/products and waste management scenarios. However, the manuscript requires further revisions and improvements before it can be considered for publication. Specific areas need clearer articulation, including the manuscript's technical details, assumptions, and policy implications. Detailed comments and suggested revisions are included below, referencing pages in the manuscript to help guide the authors in refining their work.

Response R1.0. The authors would like to thank the reviewer for their invaluable feedback on our manuscript. We agree with the reviewer's observation that certain sections, such as technical details, assumptions and policy implications, could use further context and discussion. We have carefully addressed each point below, and trust that these revisions elevate the work in the areas highlighted by the reviewer.

Comment R1.1. I suggest strengthening the conclusions to emphasize the societal and policy implications, as this could broaden the paper's impact and overall appeal. Also, I suggest providing more insight into the specific set of actions needed from industries and governments, which could improve the roadmap's importance in meeting climate goals.

Response R1.1. The authors appreciate this suggestion. The authors agree that expanding the discussion on policy mechanisms (including specific government and industry actions) that could help achieve the targets set forth in the roadmap would enhance the strength and impact of the paper. To address this point, we have added discussion around existing policies as well as potential future policies throughout the results section (P. 8 Lines 13-16, P. 11 Lines 27-33, P. 12 Lines 1-5). Examples include the US EPA's recent investment of \$275 million in solid waste infrastructure, which can help improve sorting and collection efficiency of plastic waste. Others include policy levers such as capping plastic production, requiring minimum bio-based and recycled content targets, and implementing taxes on traditional fossil-based feedstocks, which can help achieve the necessary shift towards bio-based plastics.

Comment R1.2. I also recommend expanding the discussion on technological and policy feasibility. This could be achieved by adding more detail on the economic, regulatory, and

technical challenges associated with the proposed recycling and renewable energy targets. Other suggestions would be to discuss potential policy interventions that could make the transition to bio-based plastics and increase recycling rates appear more achievable, affordable, and/or feasible in the future.

Response R1.2. The authors appreciate the reviewer for providing this feedback. The authors agree that expanding the discussion on policy scenarios, and existing technical and economic challenges would provide useful context to future readers. As such, we have added discussion of technical and regulatory roadblocks to the “pathways to net-negative emissions” section (P. 6, Lines 13-21), and we have added potential supportive policy initiatives to the roadmap section (P. 8, Lines 13-21, P. 9 Lines 4-17).

Comment R1.3 To provide a comprehensive analysis, the paper could expand the discussions of even the LCA to include other environmental impacts like water use, eutrophication, and land use changes. Addressing these alongside GHG emissions would provide a holistic perspective on bio-based plastic production’s environmental sustainability. It could also provide more insight into the trade-offs and synergies between waste management strategies and the role of conventional vs degradable bioplastics.

Response R1.3. Examining additional environmental impacts is outside the scope of this study given the level of data intensity required to run through 11 million combinations of scenarios. However, the authors agree that this is important context to add to the manuscript. Therefore, some discussion of the impacts of large-scale bio-based plastic production on land use, water consumption, and other environmental impacts has been added (P. 13 Lines 4-16). We have also added data regarding the energy intensity of each bio-based plastic to the supplemental materials as a reference (see supplemental datasheet 12). Finally, the discussion around waste management has been expanded upon to address the benefits and trade-offs of biodegradable and non-biodegradable bio-based plastics (P. 6 Lines 41-48).

Comment R1.4 There are some concerns about the terminologies used in the paper, more specifically related to misuse and lack of consistency. These need to be reviewed and adjusted accordingly.

Response R1.4. The authors have reviewed the manuscript to check for consistency and appropriateness in terminology. For example, the term bioplastic was replaced with the term bio-based plastics throughout the manuscript to be consistent and specific about the scope of the analysis. Additionally, the term carbon dioxide removal (CDR), has been replaced by the phrase net-negative GHG emissions throughout the manuscript.

Comment R1.5 I also recommend the study expand its discussion to provide clearer understanding of the technological options included in the analysis. Not only that, but they should include the potential impacts of other options not mentioned directly in this study. Most notably, the paper overlooks the role of demand changes and material substitution on the market size and, therefore, of emissions and removal potential. Also, additional routes, such as refining and processing of bio-based feedstocks, are not discussed in detail or acknowledged. This would improve the overall understanding and limitations of the paper.

Response R1.5. The authors agree with the reviewer that the discussion section of the manuscript could be expanded to address some of these points in a more cohesive manner. Therefore, additional text has been added which discusses the role of a change in plastic demand

on the outcomes of this analysis (P. 10 Lines 1-13). To provide more information regarding refining and processing of bio-based feedstocks, some additional context was added to the Methods section (P. 14 Lines 43-49, P. 15 Lines 1-19). Furthermore, a more in-depth discussion on other environmental impacts not considered in this analysis, such as land and water use, has been added (P. 13 Lines 4-16).

Minor Suggestions and Corrections

Comment R1.6. Abstract: I believe the “100,000 combinations” claim is unnecessary in the abstract, as it is unclear at this point which combinations they are and what they involve. I recommend focusing on the main takeaway messages, such as the potential and overview of the main strategies.

Response R1.6. The authors agree with the reviewer that a re-phrasing of the takeaways may improve clarity. Therefore, the abstract has been modified to include more of a discussion of the results of the roadmaps, and the minimum thresholds identified in this manuscript that are necessary for achieving net-negative GHG emission plastics (P. 1 Lines 3-17).

Comment R1.7. P2: Provide a broader overview of fossil systems. The authors acknowledge the high carbon footprint of plastics but neglect to highlight the relationship between the non-energy use of fossil fuels and the dependency this could lead to overall fossil energy. Even though the non-energy use share of crude oil can increase in the future, it is unlikely to reach very high values (above 60%), which means that plastic demand alone can determine a certain level of fossil energy production in the future. This topic is already highlighted in previous works and could improve the introduction of this paper.

Response R1.7. A brief explanation of the interdependence of the plastic sector with the fossil energy sector (including the role of fossil feedstocks in conventional plastics) has been added to the Introduction (P.2 Lines 30-36) and Discussion (P. 11 Lines 27-33, P. 12 Lines 1-5).

Comment R1.8. P2: 100% recycling is not necessary. Other studies have shown that bio-based plastics, for instance, can store carbon in long-lived applications or long-term end-of-life (such as construction materials). Additionally, conventional bioplastics can store carbon by itself if not directly incinerated or recycled (likely leading to uncontrolled end-of-life, which does not deal directly with pollution and other environmental issues, such as microplastics). Finally, other LCA studies have also shown that increased recycling rates might add emissions to the bioplastic chain, which is unlikely to happen for fossil plastics. Based on the abovementioned issues, I'm not convinced that “100% recycling” is needed to achieve CDR potential. In fact, you confirm this in the next section, when a maximum reduction was achieved with a 90% recycling rate.

Response R1.8. The authors agree with the reviewer that a recycling rate of 100% is not necessary to achieve net-zero emissions for the plastic sector (as shown in this manuscript). However, previous work that has examined the potential for net-zero or net-negative GHG emission plastics, often consider 100% recycling as a scenario (see reference 1). To avoid confusion, we have removed the “100%” and just stated that previous work found that a combination of renewable energy, recycling and bio-based plastics can lead to net-zero emissions.

- (1) Zheng, J. & Suh, S. Strategies to reduce the global carbon footprint of plastics. *Nat Clim Chang* **9**, 374–378 (2019).

Comment R1.9. P3: Some sections discuss the flexibility allowed within scenarios for achieving CDR. More explicit explanations of this flexibility (e.g., how different energy or recycling thresholds affect the outcomes) would make the results easier to interpret.

Response R1.9. The authors thank the reviewer for noting that more context for the degree of flexibility allowed and what it entails could enhance the readers' understanding. Therefore, more explanation has been added to this Results section to discuss these scenarios and provide examples (P. 3 Lines 35-39; P. 5 Lines 9-10, 25-28).

Comment R1.10. P4: Besides bioplastic share, which parameter/aspect was the most challenging (or even the determining factor) for not achieving negative emissions?

Response R1.10. Besides the bioplastic market share, another big determining factor was the make-up of the energy grid. This is in part due to the fact that some bio-based plastic production routes require significant amounts of energy (e.g. PHB production from biogas) and therefore do not provide GHG benefits compared to petroleum-based plastics unless a renewable energy grid is used. Some discussion of this has been added to the text to provide context for readers (P. 5 Lines 31-34).

Comment R1.11. P4: Expand on biomass availability and resource limitations. This section has a very brief discussion about potential limitations, but it is too broad and general. I recommend adding more specific comments, including quantitative aspects, to this discussion, which is a very important concern. Additionally, this section would welcome a more detailed discussion on technology availability concerns, such as the one originally made for one example.

Response R1.11. The authors agree with the reviewer that the availability of bio-based feedstocks is an important parameter to discuss. While there is a brief discussion of the feedstock availability within the Results section, we appreciate that more quantitative discussion would strengthen the manuscript. To address this point, a summary table has been added to the manuscript to better present this parameter (Table 1), along with some additional discussion (P. 8, Lines 39-42; P. 12 Lines 29-36, 44-49).

Comment R1.12. P4-5: Explain the role of biodegradable plastics in CDR. It is unclear how biodegradable plastics help as a CDR option. How are they able to achieve negative emissions by itself? I understand they are a very important mitigation option for conventional plastic production and end-of-life emissions.

Response R1.12. The share of biodegradable plastics is based on their technical substitution potential of today's plastics. These values are based on the report (2), which conducted a comprehensive assessment of bioplastics, considering their material performance and suitability in specific applications. Namely PLA, TPS, and PHB are often considered applicable for food packaging applications wherein recovery and recycling of plastic waste is often difficult. Given the end-of-life emissions associated with biodegradable plastics, they are only able to achieve net-negative emissions under recycling scenarios or landfill scenarios that lead to low levels of biodegradability. Therefore, while these materials may result in higher GHG emissions than non-biodegradable bioplastics, we consider them for their potential future role in applications where biodegradability is important to minimize the generation of microplastic waste and to simplify the waste treatment process. An explanation of these assumptions has been added to the text to improve clarity (P. 3 Lines 4-6).

(2) Shen, L., Haufe, J. & Patel, M. K. *Product Overview and Market Projection of Emerging Bio-Based Plastics PRO-BIP 2009 Utrecht The Netherlands*. (2009).

Comment R1.13. P4-5: Ensure consistency in terminology (1). The previous comment makes me question the usage of the term “CDR” in this paper, which uses CDR in the same as achieving global negative emissions in this sector. CDRs are options that remove carbon by themselves and, hopefully, account for a (total) net negative contribution when including all remaining positive emissions. Achieving net negative emissions does not mean that CDR is being achieved, in fact, we need CDR options to be deployed much sooner to achieve net negative emissions (or even net zero, for that matter). Please note that the two are not the same, this should be improved, and their use must be explained in the manuscript.

Response R1.13. The authors appreciate the reviewer for noting that, as written, this terminology in the manuscript could be misinterpreted. The processes modelled in this manuscript involve capturing carbon from the atmosphere (via photosynthesis) and then storing the carbon in the form of plastic products, which are then recycled, incinerated, or landfilled (or composted/anaerobically digested in the case of biodegradable plastics). Therefore, any CO₂ emissions that are not released at end-of-life (e.g., non-biodegradable plastic materials that are recycled or landfilled), are essentially removed from the atmosphere. This mechanism of CDR fits the definition reported by the IPCC. However, to avoid confusion and to ensure that the definition of CDR is properly understood, we have added additional context to the manuscript. Further, we have edited the main text to refer to these findings as net-negative GHG emission plastics, as this may be more clear, and mention the potential for plastics to act as a CDR mechanism in the discussion section (P. 9, Lines 41-45).

Comment R1.14. P4-5: Ensure consistency in terminology (2). The terms “bio-based”, “bioplastic”, and “biodegradable” are used in various contexts. Although the terminology is used correctly throughout the manuscript, a clearer distinction between these terms throughout the paper could prevent confusion for new readers, particularly in sections discussing waste management.

Response R1.14. The manuscript has been reviewed for consistency, particularly for the use of the term bio-based plastics rather than “bioplastics” throughout. Furthermore, a distinction between biodegradable and non-biodegradable plastic waste management was added to enhance clarity (P. 6, Lines 40-48).

Comment R1.15. P5: Roadmap introduction. Please note that the “large number of scenarios” was evaluated in this study, so make it more specific at the beginning of this section. From a broad context, several other scenarios could be generated from a more integrated and holistic approach, which is not the focus of this paper.

Response R1.15. The reviewer is correct in that the number of possible combinations would increase dramatically if more granularity was incorporated into the study (e.g. looking at every 1% change instead of 10% change). Therefore, the number of total combinations is not the take-away of this study, but rather the scenarios in which net-negative emissions can be achieved. The text has been adjusted to reflect this (P.3 Lines 24-25; P. 7, Lines 6-8).

Comment R1.16. P6-7: Expand on Assumptions in the Roadmap. The roadmap relies on assumptions (like a 4% annual increase in plastic demand and specific recycling rates), but it was

not clear from which set of scenarios (and with which criteria they were designed). Discussing the potential variability in these assumptions and their impact on the roadmap's feasibility could add depth to the analysis for decision/policy makers.

Response R1.16. Relevant assumptions have been added to the text to provide more context to the results of the roadmap (P. 7 Lines 1-11). Further, a sensitivity analysis was conducted to understand the impact of these assumptions on the results. A discussion of these results has been added to the text (P. 10 Lines 1-13).

Comment R1.17. P6: “Although CDR is achieved in the midterm goal”. Once again, it seems the authors are using the term CDR as the same as achieving net negative emissions. This is not accurate and needs to be fixed.

Response R1.17. The term CDR has been replaced with net-negative GHG emissions.

Comment R1.18. P6: Show more comprehensive estimates on net negative emissions or total CDR. The authors focus on one set of assumptions for the roadmap, which was already mentioned it is not clear how it was determined amongst the 100,000 scenarios. Beyond this, what is the range of negative emissions achieved in 2050 by all scenarios? What are the mean value and percentiles? Finally, how do these values compare to Integrated Assessment Model (IAM) results for the same variable and time horizon?

Response R1.18. The authors would like to thank the reviewer for this excellent point. The dataset showed a mean value of -0.47 kg CO₂e/kg (SD = 0.279), with a minimum of -1.36 kg CO₂e/kg and a maximum of -0.1 kg CO₂e/kg. We have added relevant statistics, such as the mean carbon footprint for plastics (and standard deviation), as well as average thresholds for renewable energy, bio-based plastics, and recycling, to the Results section (P. 4 Lines 27-28; P.5 Lines 5, 31-34; P. 6 Lines 3-4). While it is difficult to make direct comparisons to previous studies leveraging IAMs due to the misalignment of variables considered (e.g., the use of CCU-based plastics, which are not considered herein), we have added some discussion of other results in literature to provide context. Further, we have expanded the discussion regarding the potential reduction in GHG emissions compared to business-as-usual estimates found in previous studies (P. 9 Lines 36-39).

Comment R1.19. P8: “In this study, we find that globally, a minimum of 40% recycling, 70% renewable energy, or 60% bio-based plastics needs to be achieved to reach net-GHG-negative emissions” In here the authors use the terminology correctly, for instance. This is not the same as CDR. In fact, more CDR is needed since you still have remaining positive emissions.

Response R1.19. The authors appreciate the note (see similar response R1.13). We have modified the manuscript to consistently use the term net-GHG-negative emission bio-based plastics, rather than CDR to avoid confusion.

Comment R1.20. P8: Expand on the role of regional policies. It is not clear why the authors claim that policies at the regional level are not sufficient and why this is related to the fact that most of the demand is growing in developing countries. The connection is superfluous and generic, please expand and justify your comment.

Response R1.20. The authors appreciate the reviewer highlighting this text may contribute to confusion. The intent of this paragraph was to emphasize the global issue around plastic waste management. Even if the United States and Europe have targets for reducing plastic waste, the

majority of growth in plastic waste is expected to occur in developing countries, which may be lacking adequate waste management systems. Therefore, policies that can support and enable developing countries to also become a part of the circular bio-based plastic economy is critical. We have re-phrased this paragraph to better clarify this point (P. 12 Lines 8-9).

Comment R.21. P8-10: Address Potential Overlaps in Data Sources. In the LCA sections, some datasets for bio-based plastics appear based on similar assumptions or overlapping studies. Including a brief explanation of how these datasets complement or differ from one another would improve transparency.

Response R1.21. The life cycle inventories for the production of all nine bio-based plastics are derived from previous published work, which uses consistent scoping, allocation and impact assessment methods across all materials. Similarly, emissions associated with composting, anaerobic digestion and incineration of biodegradable bio-based plastics are also derived from a distinct previous study, where consistent calculation methods (combining stoichiometric emissions with expected biodegradation behavior from experimental findings) are used across all materials. Aside from these works, ecoinvent was utilized for LCI data for petroleum-based plastic production due to this database's extensive plastics and chemicals inventories, and we use this database for mechanical recycling inputs to maintain consistency. Further, to include chemical recycling as an end-of-life option, existing literature was leveraged for LCI data to, again, maintain consistent modeling assumptions. We appreciate the note that more detail could aid in transparency of modeling assumptions, which is always our goal. To achieve this, additional context distinguishing these datasets has been added to the Methods (P. 13, Lines 23-24, 29, 33).

Comment R1.22. P11-12: Include More Data on Specific Feedstock Availability. The discussion on bio-based feedstock availability should be expanded with quantitative estimates. This would give readers a better sense of the practicality of meeting bio-based plastic targets based on global resource limits and sustainability concerns about the supply chain. Additionally, how can land-use change, and agricultural (AFOLU) emissions affect the GHG emissions results from plastics? If accounted in other sectors, due to inventory guidelines, how would this affect overall achievement of net negative emissions?

Response R1.22. The authors agree with the reviewer that feedstock availability is an important factor to discuss in the manuscript. To present this quantitatively, a summary table has been added to the results section to highlight the amount of feedstocks required to meet the goals outlined in the roadmap, compared to currently available resources (Table 1). In this analysis, direct land-use change emissions (e.g. emissions associated with converting the necessary land required for crop production), are considered. However, indirect land-use change emissions, which capture shifts in land-use as a result of changes in interrelated markets, are outside the scope of this study. To improve clarity around these points, context of the potential impacts of indirect land-use change have been added to the Discussion (P. 13 Lines 4-15).

Comment R1.23. P13: Role of multi-product routes. The study focused on plastic production options, which, from my understanding, does not include technological options that are able to make a wider range of products (including energy and non-energy products) from biobased sources. These could have the benefit of decarbonizing one sector while simultaneously

decoupling fossil dependency for chemical feedstock. This should be acknowledged, as they were the focus of recent articles in the same journal.

Response R1.23. The authors appreciate the reviewer for bringing up this point. The authors agree that optimizing biorefineries to coproduce energy fuels, or other value-added chemical products, in addition to bio-based plastics, can help optimize profits and reduce impacts. Therefore, some discussion regarding the potential for co-product valorization was added to the text (P. 12 Lines 44-49).

Reviewer #2

Comment R2.0 The paper applies a LCA-based method to calculate pathways for plastics production that lead to carbon dioxide removal as a strategy for climate change mitigation. It considers several biodegradable and non-biodegradable alternatives for fossil-based plastics as well as their production, EoL emissions, feedstock availability, renewable energy use, and market penetration to understand under which conditions the global plastic industry would become net-negative. The topic is interesting and timely to support climate policy for the plastics production industry. First of all, I would like to thank the authors for the easy and interesting read. The data provided allows for validation of results and is relevant to support future studies. I have a few comments below before accepting it for publication, which are I hope are useful to improve the key argument made in the paper.

Response R2.0. The authors would like to thank the reviewer for providing such valuable feedback. We are grateful for their time and consideration. We also appreciate the thoughtful comments and suggestions that help to strengthen the impact of this paper. We have addressed each point carefully, as detailed below.

Comment R2.1. A key limitation of the study concerns the temporal dynamics of plastic production, use, and waste generation, i.e. availability for chemical or mechanical recycling. The authors assume that “the quantity of plastics reaching end-of-life each year is equal to the quantity of plastics being produced”, “to have a “worst-case scenario” estimate for end-of-life emissions.” On one hand, it is a worst-case scenario for emissions, but on the other hand, it is the best-case scenario for plastic waste availability to be recycled. How does this affects current results and how this could be addressed in future studies?

Response R2.1. The reviewer makes an excellent point regarding the temporal limitations of this study. However, the assumption made herein that the amount of plastic reaching end-of-life is equal to the amount being produced, still provides a “worse case scenario” estimate in terms of net-emissions. This output is because, when a material is assumed to be recycled in this analysis, no further carbon dioxide storage is associated with it. Meaning that the more plastics that are recycled, the less plastics are produced, and thus less annual biogenic resources going into the carbon backbone of the plastics (and hence additional carbon dioxide storage) can be achieved. Similarly, scenarios with low recycling rates (and therefore high incineration rates) are going to increase emissions. However, the authors agree with the reviewer that this is an important limitation to address in the manuscript, therefore some context has been added to discuss the impacts of this assumption (P. 10 Lines 13-22).

Comment R2.2. What precisely makes bio-based non-biodegradable plastics become a carbon

sink? Previous studies emphasize CDR role of bioplastics in long-lived applications, but this does not seem to be the case here. Is it the continuous recirculation of biogenic carbon through mechanical/chemical recycling?

Response R2.2. Yes, the reviewer is correct. In this case, it is the repeated recycling and recirculation of these bio-based, non-biodegradable plastics that enable them to act as carbon storage. However, to address concerns around the use of the term “carbon dioxide removal” in this study, we have edited the text to refer to these materials as being “net-negative GHG emission” plastics, rather than carbon sink plastics.

Comment R2.3. Assumptions about future demand rely on the historical annual 4% growth. At the same time, demand increase is precisely what drives CDR and net negative emissions in the plastics industry in this study. How would the results change considering lower rates of demand increase (e.g., due to bans on single-use plastic taking off)?

Response R2.3. The authors agree with the reviewer that considering different growth rates is important for this analysis. To provide this additional context, we conducted a sensitivity analysis. Namely, the results of the roadmap were re-examined assuming no future increase in plastic demand, a 2.5% annual growth rate, and an 8% annual growth rate. We have added some discussion of these results to the text (P. 10, Lines 1-13), and the full results can be found in the supplemental materials, data sheet 13.

Comment R2.4. Adding to the comment above, circularity is mentioned more than once in the manuscript. However, only lower ranked R-strategies are being discussed in the study (recycling). Shouldn't a broader circular bioeconomy strategy for the plastics industry also consider reducing plastic demand and extending the lifetimes of plastic products?

Response R2.4. The reviewer brings up an excellent point that reducing demand and/or extending the lifetime of plastic products is an important parameter to consider for achieving global sustainability. While the focus of this paper was on the potential for the plastic sector to act as a pathway for carbon storage, we recognize that in order to conserve resources and energy, creating a more circular economy and reducing plastic demand is crucial. Therefore, we have included some discussion on the importance of such measures and include references to studies that have looked at the impacts of such measures (P. 10 Lines 6-7).

Comment R2.5. Figure 1: That is an interesting figure but could be improved for readability. Consider: 1) moving the letters a, b, c, d to the upper left side of each panel; 2) bringing non-biodegradable/biodegradable to the left of the labels in the y axis (I'm assuming there are valid for all the panels?); 3) in general, labelling the three main categories there would facilitate for the reader, there is a lot of info in that figure; 4) why in scenario A there is an outlier with 60% renewable energy? Similarly, why is there an outlier on 60% incineration in scenario C? 5) which year is this figure referring to? Or is static considering current production?

Response R2.5. The authors appreciate the reviewer for providing detailed feedback for Figure 1. We have implemented the suggested changes. Further, we have specified the meaning of the outliers in the figure caption, as well as provided context for the year of analysis (in this case, results are showing what strategies would need to be implemented to achieve negative emissions on a per-kg plastic basis, and, therefore, does not account for scaling or demand of plastics).

Comment R2.6. On the roadmap: the authors presented net emissions until 2050. What does it

mean in terms of renewable energy use (for plastics production and recycling) and feedstock use in EJ/yr? What about land-use? These values would help clarifying the role of the sector in competition with others for renewable resources globally.

Response R2.6. The primary focus of this manuscript is to examine the GHG emissions associated with the plastic sector and the potential to transition to bio-based plastics as a mitigation strategy for GHGs. With that being said, we agree with the reviewer that energy use and land-use are crucial topics to consider when examining the potential for a bio-based economy. Therefore, we have added data to the supplemental materials regarding the energy intensity of the bio-based plastic production routes considered herein (supplemental data sheet 12). It is important to note that the roadmap scenarios only consider the use of second and third-generation feedstocks (or non-edible biomass) for bio-based plastic production. Not only does this consideration limit competition with food, but it also provides a value chain for an otherwise low-valued residue (e.g., corn stover, wheat straw). Given that these feedstocks are byproducts of food production, our scope does not consider them as main drivers of land-use change. Due to this complication regarding land-use allocation, along with the uncertainty of land-use change emissions is high, we believe that a more thorough analysis of land-use change impacts would be beneficial to consider in future work. However, to address this concern, we have also added some discussion, including references, to comprehensive studies on land-use change impacts associated with bio-based plastics (P. 13, Lines 4-11).

Comment R2.7. The main message that I take from this study, given the limitations on future plastic demand, uses and their temporal dynamics, is that the key to net-negative plastics sector lies on producing more plastics, transitioning towards bio-based feedstocks, and improving collection, sorting and recycling methods. If we aim for the thresholds mentioned regarding renewable energy, recycling and feedstock, the plastics industry will reach net-negative emissions. Is that the main message of the paper?

Response R2.7. The authors appreciate the reviewer for providing their interpretation of the results. The reviewer is correct in their assumption that producing more plastics made from bio-based materials, along with implementing a closed loop recycling system and renewable energy grid, could ultimately lead to greater magnitudes of carbon storage. However, given that a comprehensive set of environmental impact categories are not considered herein, it is important to keep in mind that there could be trade-offs associated with increased bio-based plastic production such as land-use change and energy demand, as discussed in Comment 2.6. We have added some discussion to the results to emphasize this point (P. 10, Lines 6-13).

Comment R2.8. I think the discussion section can be much improved. The data collection and modelling in the micro-/technology level is impressive and very comprehensive, but I am missing a discussion that connects these results with global climate policy. For instance, what these results mean for global fossil fuels phase-out? What are the implications for achieving global climate agreements?

Response R2.8. The authors appreciate the reviewer for their thoughtful suggestion. The authors agree that expanding the discussion around global policy initiatives and roadblocks for the phase out of fossil-based plastics would strengthen the paper. Therefore, additional policy levers have been added throughout the Results section to support the targets identified in the roadmap (P. 8 Lines 13-21; P. 9 Lines 4-17). In addition, some discussion around the interdependencies of the plastic sector with the fossil-fuel industry, and the associated carbon lock-ins, have been added to

the Introduction and Discussion (P. 2 Lines 30-36; P. 11 Lines 27-33, P. 12 Lines 1-5), as well as some discussion of how these results relate to global climate targets (P. 9 Lines 41-45).

Minor:

Comment R2.9. L48: There's a 'that' missing before 'to achieve net-zero emissions

Response R2.9. This typographical error has been corrected.

Comment R2.10. L50: 58% reduction relative to which scenario/year? The whole sentence is a bit confusing, consider rewriting.

Response R2.10. This sentence has been modified to state that a 58% reduction in GHG emissions compared to current GHG emissions from the petroleum-based plastic sector, could be achieved in the near term (P. 2 Lines 14-17).

Comment R2.11. L201: Same as above (why the difference in 57% and 58% reductions?)

Response R2.11. This issue has been corrected.

Comment R2.12. L217: 25-58%? Is this correct?

Response R2.12. The references have been reviewed, and the numbers have been corrected. Both statistics come from a European plastics report that includes various future projections. To be consistent the numbers have been corrected to reflect the range of the baseline scenario (25-27% recycling).

Comment R2.13. L416: 0.1 kg CO₂e/kg (kg before CO₂ is missing)

Response R2.13. This typographical error has been corrected.

SI file:

Comment R2.14. L66: bio-based feedstocks?

Response R2.14. This typographical error has been corrected.

Title: Using biogenic resources to make plastics a global carbon sink: a roadmap for sustainable decarbonization by 2050

Manuscript ID: NCOMMS-24-58695

Response to Reviewer Comments

Comment 0. Thank you to the authors for the revisions, which have significantly improved the readability, content, overview of limitations, and overall understanding of the paper. The manuscript is much stronger than the initial version, thanks to the additional information provided. However, I still have a few minor corrections and clarifications before it can be accepted for publication. Below, I refer to pages and lines as indicated in the document with track changes.

Response 0. The authors would like to thank the reviewers for their support and for taking the time to provide thoughtful feedback on our work. We have provided point-by-point responses to the comments presented below. Please note that the section and line numbers in our responses refer to the revised manuscript with tracked changes.

Comment 1. Table 1 is a valuable addition, but it is not referenced in the text. It would be useful to briefly discuss the values for cooking oil beyond what is mentioned in P17L22-23. While the benefits of co-production are acknowledged, the scale of production raises feasibility concerns regarding the proposed roadmap. Consider discussing alternatives to cooking oil/rapeseed oil for Bio-PP or other strategies to enhance feasibility. For instance, how much of other fuels are co-produced, and how does this compare to the SAF demand expected for 2030/2050?

Response 1. The authors appreciate the reviewer for highlighting this point. We have specifically referenced Table 1 in the text (P. 7 Lines 11-13). In addition, given the restraints on used vegetable oil, we have added considerable text to the results section to discuss alternative feedstocks and production routes for bio-PP (P. 9 Lines 30-38).

Comment 2. The study considers wind electricity and biogas as renewable energy sources. Table 1 appears to focus on bio-based feedstocks, but total electricity demand is unclear. Many studies emphasize the increasing electricity requirements for net-zero plastics/chemicals, particularly for CCU pathways. For example, Kätelhön et al. (2019) (<https://doi.org/10.1073/pnas.1821029116>) estimates that shifting chemical feedstocks to carbon capture and utilization would require low-carbon electricity equivalent to 55% of globally projected generation in 2030 (not just renewables!). How does this study compare, given its focus on bio-based alternatives?

Response 2. The authors appreciate the reviewer for highlighting this point. Although we report the energy demand for each plastic in the supplemental materials, we agree that including these metrics in Table 1 would be beneficial to the reader. Therefore, we have added two columns to Table 1 to report the total energy demand (including both electricity and heat) required for the plastic production scenarios highlighted in the roadmap, as well as the percent of total global energy production required. The energy demand values include the impacts of bio-based plastic production, as well the remaining fossil-based plastic production for each scenario (e.g. in 2030, 40% bio-based plastics, 60% fossil-based plastics). Further, we calculate the energy demand for a business-as-usual approach, in which all plastics are fossil-based and are assumed to have an energy-demand of 32 MJ/kg. These results are provided in the Supplementary Data Sheet 12. In addition, we expand the results section to include discussions of energy demand associated with each of the roadmap scenarios (2030,2040 and 2050) (P. 8 Lines 27-28; P.9 Lines 38-42,47-48; P. 12 Lines 19-23). We also agree with the reviewers point that CCU-based plastic production often requires high energy demand. Therefore, we have added some discussion to the text comparing the energy demand of CCU technologies to bio-based plastic production routes, including the reference provided by the reviewer (P. 7, Lines 24-29).

Comment 3. The criteria for selecting net-negative GHG scenarios (based on LCI and kg plastic) and their alignment with the global roadmap remain unclear. The authors mention TRL as a proxy for short-, medium-, and long-term implementation, as well as resource availability (P7L8) and an assumption of 100% renewable energy (P16L38). A sensitivity analysis was added for demand, yet the decision-making process behind scenario selection is still not entirely transparent or scattered in multiple subsections. Moreover, the first sentence of the discussion also mention the minimum shares of recycling, renewable energy, and bio-based feedstock for net negative emissions in plastics (per kg of plastics). The next sentence mention the required shares for the roadmap, which give me an indication that those are the minimum shares for net negative, considering TRL and resource constraints. However, the supplementary material (SDS11) also includes a column labeled "carbon optimal pathway." While this terminology is used in other studies, there is no clear indication of an optimization method applied to determine the carbon-optimal long-term pathway in this study. Is it the minimum effort for net-negative global emissions? Maximum cumulative net negative emissions? Consider adding a sentence, a scheme, or short paragraph that summarizes the steps taken to define the roadmap.

Response 3. The authors agree with the reviewer that the introductory paragraph for the roadmap results could use more information regarding how the specific scenarios were chosen. To clarify, we provided more detailed information about how TRL levels were leveraged, alongside resource availability and relevant policies, to determine the specific scenarios (P. 6 Lines 28-43). We also clarify that the roadmap does not directly use the minimum thresholds identified in the previous paragraphs, but rather a combination of strategies that are technologically feasible. In

addition, we have added a clarifying statement to Supplementary Data Sheet 11, which defines what is meant by carbon optimal pathways. The carbon optimal pathway column refers to the “best case scenario” resulting in lowest possible GHG emissions in each study. For this work, these best options correspond to the 2050 scenario in the roadmap (utilizing 100% renewable energy, 90% bio-based plastics, and 69% recycling rate). Limits for recycling in this “best case scenario” are based on the assumptions that 10% of plastic waste will always be lost to landfills or the environment, and that biodegradable plastics, which are assumed to make up 21% of the market in 2050, will be treated via composting and anaerobic digestion.

“While the previous paragraphs highlight the minimum thresholds required for recycling, bio-based plastics, and renewable energy, here we provide a roadmap which leverages a combination of these solutions to achieve net-negative emissions by 2050. The combination of solutions presented for the short term (2030), medium term (2040) and long term (2050) were chosen based on TRLs, resource availability, and relevant policies (Fig 2). TRLs are based on the United States Department of Agriculture definitions ⁴⁰, and resource availability data are from the Food and Agricultural Organization 2020 statistics ⁴¹. For the near term (2030) scenario, only bio-based plastics with a TRL level of 9 are considered. The 2040 scenario considers bio-based plastics with TRL levels of 5 and above, which encompasses all of the bio-based plastics herein except for bio-PP as it is not considered feasible in the medium term due to resource constraints (see Table 1). While mechanical recycling, composting and anaerobic digestion all have TRL levels of 9, we base their implementation on relevant policies (as described below). Similarly, given the lack of robust infrastructure for collecting, sorting and treating biodegradable plastics, full implementation of composting and anaerobic digestion for these materials is not considered feasible until the medium term (2040). The magnitude of emissions and uptake are calculated based on the assumption that plastic demand will continue to grow at an annual rate of 4% per year. While these strategies do not need to be implemented at this scale or in this order, this analysis suggests that pathways to net-uptake in the plastics industry are feasible within the coming decades.”